# *Diff-ICMH*: Harmonizing Machine and Human Vision in Image Compression with Generative Prior

Ruoyu Feng[1*]    Yunpeng Qi[1*]    Jinming Liu[2]    Yixin Gao[1]
Xin Li[1†]    Xin Jin[2]    Zhibo Chen[1†]

[1]University of Science and Technology of China
[2]Eastern Institute of Technology, Ningbo

## Abstract

Image compression methods are usually optimized isolatedly for human perception or machine analysis tasks. We reveal fundamental commonalities between these objectives: preserving accurate semantic information is paramount, as it directly dictates the integrity of critical information for intelligent tasks and aids human understanding. Concurrently, enhanced perceptual quality not only improves visual appeal but also, by ensuring realistic image distributions, benefits semantic feature extraction for machine tasks. Based on this insight, we propose *Diff-ICMH*, a generative image compression framework aiming for harmonizing machine and human vision in image compression. It ensures perceptual realism by leveraging generative priors and simultaneously guarantees semantic fidelity through the incorporation of *Semantic Consistency loss (SC loss)* during training. Additionally, we introduce the *Tag Guidance Module (TGM)* that leverages highly semantic image-level tags to stimulate the pre-trained diffusion model's generative capabilities, requiring minimal additional bit rates. Consequently, Diff-ICMH supports multiple intelligent tasks through a single codec and bitstream without any task-specific adaptation, while preserving high-quality visual experience for human perception. Extensive experimental results demonstrate Diff-ICMH's superiority and generalizability across diverse tasks, while maintaining visual appeal for human perception.

## 1 Introduction

The digital era has driven an explosive growth in network data. Compression algorithms are crucial in the storage and transmission. Image compression, a cornerstone of visual signal processing, is essential for both technological advancements and the operational efficiency of numerous digital systems. In terms of algorithmic development, traditional compression standards such as JPEG [1], JPEG2000 [2], H.264/AVC [3], H.265/HEVC [4], and H.266/VVC [5] have been widely applied. More recently, learned image compression methods [6–23] have emerged, demonstrating remarkable performance. However, these methods are primarily optimized for human visual perception. Concurrently, the rapid development of artificial intelligence technology means that the scale of visual data consumed by downstream intelligent tasks is growing substantially. This shift in application scenarios highlights the urgent need for developing compression methods tailored for intelligent tasks.

Representative existing approaches to image compression for machines (ICM) are depicted in Fig. 1 (a)-(c). Traditional codec-based methods optimize compression for machine tasks through quantization parameter tuning [24–29], strategic bit allocation [30–33], or by integrating neural network-based pre/post-processing modules [34–37]. While these approaches enable a single codec

---

*Equal contribution.
†Corresponding authors.

39th Conference on Neural Information Processing Systems (NeurIPS 2025).

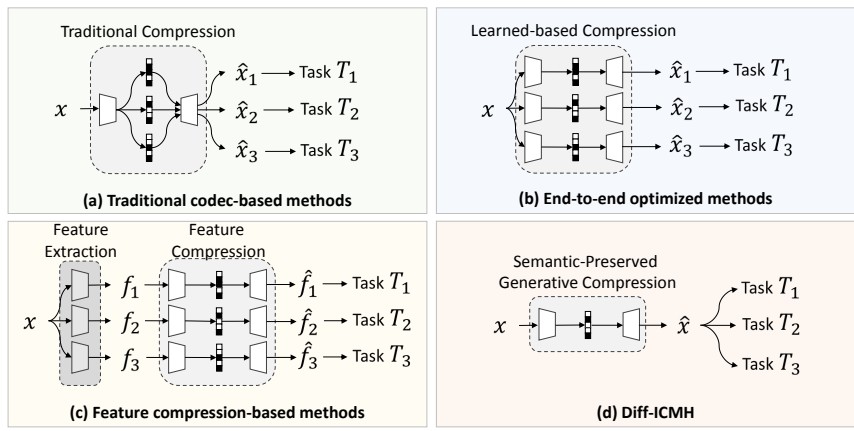

Figure 1: Comparison of compression and reconstruction workflows between different methods.

to support multiple tasks, their generalization and performance are often constrained due to the codec's inherent fidelity-oriented design and its non-differentiable nature. Task-driven end-to-end optimized methods [38–45] typically achieve superior performance on specific tasks but exhibit limited generalization at both the bitstream and codec levels. Feature compression-based methods [46–61] directly compress intermediate features from *specific* task models, often yielding a more favorable rate-distortion trade-off, while also reducing the computational burden on the server-side. However, such methods still face generalization challenges across different intelligent task models.

Optimizing image compression solely for specific intelligent tasks limits generalization across diverse tasks and human perception. To overcome this, we introduce Diff-ICMH (Fig. 1 (d)), a conditional generation approach targeting both versatile machine task support and human visual quality. From a novel perspective, we identify that performance degradation in intelligent tasks using compressed images primarily stems from two core information losses: (1) compromised *semantic integrity* (loss of core intelligible meaning), which directly impairs task analytics; and (2) reduced *perceptual realism* (textures and details deviating from natural distributions), leading to distribution mismatches that hinder feature extraction and cause inaccurate semantic analysis. Diff-ICMH tackles these issues by first employing a diffusion model-based generative framework, leveraging pre-trained models like Stable Diffusion [62], to ensure perceptual realism and mitigate domain shifts. Crucially, to preserve semantic integrity, we introduce Semantic Consistency loss (SC loss), which aligns features extracted by the pre-trained diffusion models. Furthermore, a Tag Guidance Module (TGM) utilizes efficiently coded, word-level image tags to activate generative priors, thereby enhancing both the subjective quality and semantic clarity of the reconstructed images.

The contributions of this paper are as follows:

- We introduce an innovative perspective for designing a versatile codec that jointly serves multiple intelligent tasks and human visual perception, identifying semantic fidelity and perceptual realism as critical determinants for this unification.

- Building on this insight, we introduce Diff-ICMH, which integrates generative priors with robust semantic information preservation, realized through proposed Semantic Consistency loss and Tag Guidance Module.

- Extensive experiments validate our approach, showcasing state-of-the-art performance across 10 diverse downstream intelligent tasks. All results are achieved without task-specific adaptation training, while concurrently delivering high-quality reconstructions for human visual perception.

## 2 Related work

### 2.1 Image compression for machines

Research in image compression for machines (ICM) has predominantly explored three primary directions. First, traditional codec-based methods adapt standardized formats like JPEG [1] and

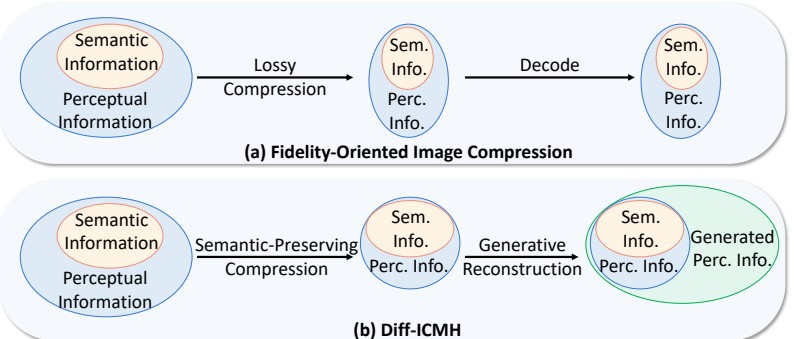

Figure 2: Illustration of the information transformation during compression processes in signal fidelity-oriented image compression and Diff-ICMH.

H.265 [4] by optimizing quantization parameters [24–29], employing strategic bit allocation [30–33], or applying neural network pre/post-processing [34–37]. While these methods offer the advantage of supporting multiple tasks with a single codec, their inherent signal fidelity-oriented design often limits their generalization capabilities and can lead to performance bottlenecks. Task-driven end-to-end optimized methods utilize learned codecs to directly optimize for specific machine tasks [38–43]. However, task-driven optimization often leads to poor generalization across different tasks. To mitigate this limitation, recent works [44, 45, 43] implement adaptation mechanisms aimed at enabling efficient task adaptation with minimal trainable parameters in the codec. Feature compression methods [46–61] directly encode intermediate neural network features instead of images, offering efficiency gains especially in cloud-edge scenarios, though they remain tightly coupled with specific feature extractors and cannot support human viewing.

## 2.2 Generative image compression

The basic objective of lossy compression is to optimize the trade-off between bit rate and quality. However, because traditional fidelity-oriented metrics like Peak Signal-to-Noise Ratio (PSNR) often correlate poorly with human visual perception, generative codecs have emerged as a promising direction for enhancing perceptual quality. Foundational theoretical work on rate-distortion-perception relationships [63–65] has catalyzed practical advances in this field. Two predominant approaches have gained prominence. First, GAN-based [66] methods [67–72] typically employ adversarial training to enhance perceptual quality. Second, diffusion model-based approaches [73–83] optimize for rate, distortion, and generation quality, either by training models from scratch or by leveraging large pre-trained models like Stable Diffusion [62]. Both approaches can be seen as using decoded features as conditional signals to guide perceptually realistic reconstructions. The ongoing advancement of generative models holds significant promise for their integration into image compression, paving the way for high-fidelity visual reconstructions at low bit rates.

## 3  Motivation

As shown in Figure 2 (a), traditional signal fidelity-oriented image compression methods [1–23] typically focus on indiscriminately minimizing pixel-wise errors, such as Mean Squared Error (MSE). This indiscriminate optimization, however, often incurs both semantic distortion and pixel-level perceptual mismatch. Semantic distortion directly impacts the information completeness crucial for downstream tasks. Furthermore, perceptual mismatch manifests as domain shifts, leading to error propagation during feature extraction and hindering the accurate mapping of inputs to semantic features. Generative image compression, by its very nature of reconstructing images to mimic natural distributions, is inherently well-suited to mitigating such domain shifts.

To investigate this, we compared feature divergence (1 minus cosine similarity) between fidelity-oriented codecs (VTM-18.2, ELIC) and a GAN-based generative one (MS-ILLM [72]). As illustrated in Figure 3, while fidelity-oriented codecs exhibited smaller feature differences at shallow network layers (stem layer) due to superior signal fidelity (higher PSNR), they suffered significantly larger divergence at deeper layers (layer2 and layer4). This confirms that the realistic textures produced

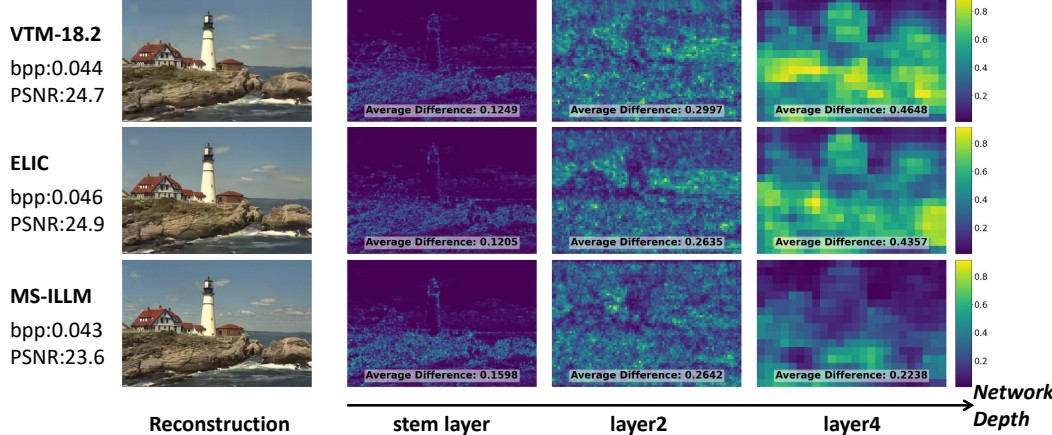

Figure 3: Variation in the difference (1 minus cosine similarity) between features extracted from reconstructed images by different codecs and those from original images when input to pre-trained ResNet50 [84], shown against increasing network depth.

by generative image compression effectively mitigate error accumulation arising from domain shift. However, realistic textures alone are insufficient for optimal intelligent task performance if semantic information integrity is compromised. Accurate extracted representations require *both* realistic textures and complete semantic information.

Diff-ICMH addresses these dual challenges based on a novel design philosophy: robust semantic information is prioritized and preserved during encoding, which then guides a generative reconstruction of perceptually realistic details during decoding, illustrated in Figure 2 (b). We realize this by developing a generative compression method leveraging a pre-trained Stable Diffusion model [62], further enhanced by our proposed Semantic Consistency loss (SC loss) and Tag Guidance Module (TGM). The subsequent sections will detail the overall framework of Diff-ICMH, along with the specifics of the SC loss and the TGM.

## 4 Diff-ICMH

### 4.1 Overall framework

The Diff-ICMH framework is depicted in Figure 4. The input image $\mathbf{x}$ is compressed and reconstructed as a latent feature $\hat{\mathbf{z}}$, targeting the VAE latent space of the pre-trained Stable Diffusion model. Concurrently, the tag extractor derives word-level tags $\mathbf{c}$ from $\mathbf{x}$ to capture coarse-grained semantics. In practical usage, the bitstream contains the compressed latent variables and the extracted tag IDs.

A key design choice in Diff-ICMH is the decoding of the bitstream directly into the VAE latent space of the pre-trained diffusion model instead of pixel space. This is motivated by several compelling properties of the latent space. It is inherently optimized for feature-level perceptual quality and realism, a result of its training objectives that include perceptual-oriented LPIPS loss [85] and adversarial losses [66]. Furthermore, this space provides a compact and perceptually rich representation (e.g., $8 \times 8$ spatial downsampling in Stable Diffusion) which effectively filters semantically irrelevant redundancy from the pixel domain. Consequently, optimizing for fidelity within this latent space enables the bitstream to prioritize perceptually salient and semantically coherent information, crucial for robust performance in downstream machine tasks.

Generative reconstruction is then conducted with $\hat{\mathbf{z}}$ as condition. Specifically, $\hat{\mathbf{z}}$ is fed to the control module, while the noisy latent $\mathbf{z}_t$ (at timestep $t$) is input to the diffusion model. The control module adapts ControlNet [86]-like architecture for conditioning on the downsampled latent feature $\hat{\mathbf{z}}$. Subsequently, generative reconstruction is performed using the control module and the pre-trained Stable Diffusion model. The reconstructed feature $\hat{\mathbf{z}}$ and the latent feature $\mathbf{z}_t$ at timestep $t$ are input to the control module and diffusion model, respectively. The diffusion model then predicts the noise $\boldsymbol{\epsilon}_\theta(\mathbf{z}_t, \hat{\mathbf{z}}, \mathbf{c}, t)$. Following the standard reverse diffusion process for $T$ steps, we obtain the denoised

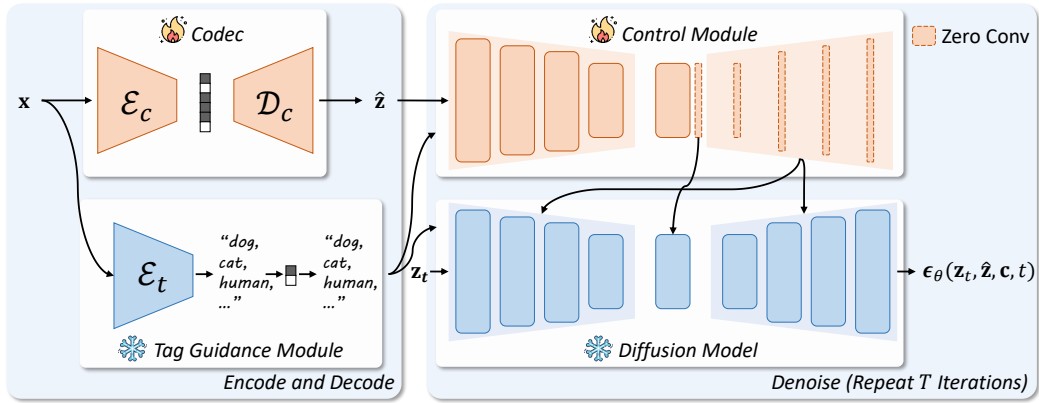

Figure 4: **Overview of Diff-ICMH.** Diff-ICMH consists of two parts: (left) image encoding/decoding and tag extraction; (right) condition-based image reconstruction. For simplicity, skip connections are omitted in the diagram.

latent $\mathbf{z}_0$. This $\mathbf{z}_0$ is then passed through the VAE decoder $\mathcal{D}(\cdot)$ to yield the final reconstructed image $\hat{\mathbf{x}}$. See Appendix for the detailed architecture of control module.

## 4.2   Semantic Consistency loss

To effectively preserve crucial semantic information during lossy coding, which is often compromised in traditional approaches, we propose the Semantic Consistency loss (SC loss). The core idea is to enforce semantic alignment between representations derived from the ground truth and the decoded signal. This is achieved by framing it as a pretext task: both signals are projected into a shared, high-quality semantic space where their representations are encouraged to be consistent.

The choice of this semantic space is critical for ensuring that the preserved semantics are both rich and generalizable for diverse downstream applications. Recent works [87–92] have demonstrated that large-scale pre-trained diffusion models possess strong inherent capabilities for image understanding and semantic feature extraction. Inspired by this, we leverage features extracted by these pre-trained diffusion models to instantiate our semantic space and guide the SC loss.

Specifically, as shown in Figure 5 (a), the ground truth latent variable $\mathbf{z} = \mathcal{E}(\mathbf{x})$ and the decoded feature $\hat{\mathbf{z}} = \mathcal{D}_c(\mathcal{E}_c(\mathbf{x}))$ are separately input into the pre-trained diffusion model, where $\mathcal{E}$ indicates the VAE's encoder and $\mathcal{E}_c, \mathcal{D}_c$ corresponds the the codec's encoder and decoder. We utilize the model's forward propagation $f(\cdot)$ as a bridge mapping from the original latent space to the semantic space, and align the two in this semantic space, optimizing the encoder-decoder through backpropagation.

This alignment is enforced by maximizing the similarity between the semantic representations $f(\mathbf{z})$ and $f(\hat{\mathbf{z}})$. The SC loss, $\mathcal{L}_{\text{sem}}$, is therefore formulated as:

$$\mathcal{L}_{\text{sem}} = -\mathbb{E}_{\mathbf{z},\hat{\mathbf{z}}} \left[ \frac{1}{N} \sum_{n=1}^{N} \text{sim}(f(\mathbf{z})_n, f(\hat{\mathbf{z}})_n) \right], \tag{1}$$

where $N$ represents the number of spatial positions in the feature, $n$ indicates the spatial position index of the feature, and $\text{sim}(\cdot, \cdot)$ is a predefined similarity measurement function. In this paper, we instantiate the $\text{sim}(\cdot, \cdot)$ function using cosine similarity:

$$\text{sim}(\mathbf{z}, \hat{\mathbf{z}}) = \frac{\mathbf{z}^T \hat{\mathbf{z}}}{|\mathbf{z}|_2 |\hat{\mathbf{z}}|_2}. \tag{2}$$

## 4.3   Tag Guidance Module

To activate generative priors in pre-trained diffusion models with minimal bitrate overhead, we introduce the Tag Guidance Module (TGM). As depicted in Figure 5 (b), a pre-trained tag extractor $\mathcal{E}_t$ (e.g., Recognize Anything [93] is used in this paper) first generates instance-level semantic tags for the

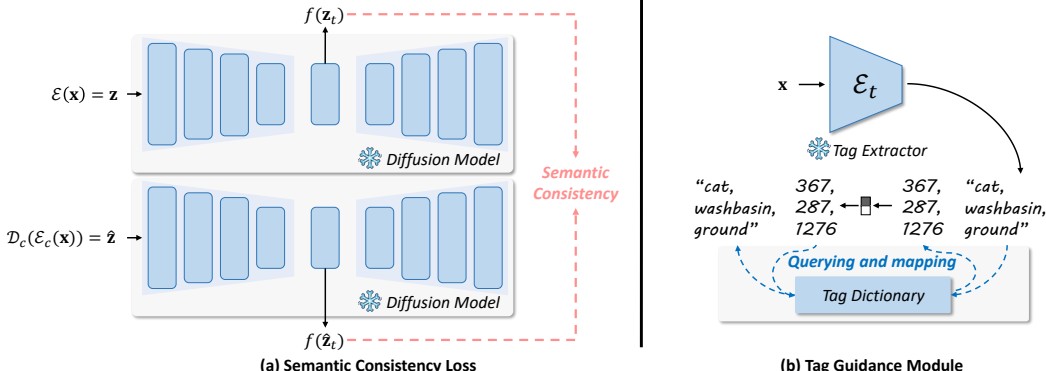

Figure 5: **(a) Semantic Consistency loss.** The latent variable $\mathbf{z}$ and the decoded latent variable $\hat{\mathbf{z}}$ are projected through the pre-trained diffusion model, resulting semantic representations that are then aligned. **(b) Tag Guidance Module.** Tags are extracted via tag extractor and losslessly compressed.

input image $\mathbf{x}$. These tags are subsequently mapped to numerical indices using a predefined dictionary and then losslessly encoded. At the decoder, these indices are converted back to their corresponding word-level tags via the dictionary. Finally, these tags are formatted as comma-separated text strings and input as conditioning to both the diffusion model and the control module.

This tag-based guidance mechanism incurs a very low bitrate overhead (approximately 100 bits per image) due to the typically small number of tags per image and a compact predefined dictionary. We therefore employ simple fixed-length coding for the tag indices. During the inference stage, Classifier-Free Guidance (CFG) [94] is utilized with these text tags to steer the diffusion model towards generating reconstructions with more distinct and accurate semantic content.

## 4.4 Loss function

Our final optimization objective, $\mathcal{L}_{\text{final}}$, combines four components: a rate loss $\mathcal{L}_{\text{rate}}$, a latent space reconstruction loss $\mathcal{L}_{\text{dist}}$, a diffusion model noise prediction loss $\mathcal{L}_{\text{diff}}$, and the SC loss $\mathcal{L}_{\text{sem}}$.

The rate loss $\mathcal{L}_{\text{rate}}$ penalizes the estimated entropy of the quantized primary latent variables $\hat{\mathbf{y}}$ and the quantized hyperprior latents $\hat{\mathbf{z}}_h$, following previous methods [7]:

$$\mathcal{L}_{\text{rate}} = \mathcal{R}(\hat{\mathbf{y}}) + \mathcal{R}(\hat{\mathbf{z}}_h). \tag{3}$$

During training, quantization is approximated by adding uniform noise [6] and we use the straight through estimator for the input of synthesizer.

The latent space reconstruction loss $\mathcal{L}_{\text{dist}}$ measures the distortion between the target VAE latent $\mathbf{z} = \mathcal{E}_{\text{VAE}}(\mathbf{x})$ and reconstructed latent $\hat{\mathbf{z}} = \mathcal{D}_c(\hat{\mathbf{y}})$. We define it as the Mean Squared Error (MSE):

$$\mathcal{L}_{\text{dist}} = \|\mathbf{z} - \hat{\mathbf{z}}\|_2^2 = \|\mathcal{E}_{\text{VAE}}(\mathbf{x}) - \mathcal{D}_c(\hat{\mathbf{y}})\|_2^2, \tag{4}$$

where $\mathcal{E}_{\text{VAE}}$ is the encoder of the pre-trained VAE (associated with the diffusion model) and $\mathcal{D}_c$ is the codec's decoder operating on $\hat{\mathbf{y}}$.

The diffusion model's noise prediction loss $\mathcal{L}_{\text{diff}}$ follows the standard formulation:

$$\mathcal{L}_{\text{diff}} = \mathbb{E}_{\mathbf{z},t,\mathbf{c},\boldsymbol{\epsilon}\sim\mathcal{N}(\mathbf{0},\mathbf{I})} \left[ \|\boldsymbol{\epsilon} - \boldsymbol{\epsilon}_\theta(\mathbf{z}_t, \hat{\mathbf{z}}, \mathbf{c}, t)\|_2^2 \right], \tag{5}$$

where $\mathbf{z}$ is the clean VAE latent (serving as $\mathbf{z}_0$ for the diffusion process), $\mathbf{z}_t$ is its noised version at timestep $t$, $\boldsymbol{\epsilon}$ is the sampled Gaussian noise, $\boldsymbol{\epsilon}_\theta$ is the network's predicted noise, $\hat{\mathbf{z}}$ is the reconstructed latent from our codec acting as a condition, and $\mathbf{c}$ represents tags from TGM.

The final composite loss function is a weighted sum of these components:

$$\mathcal{L}_{\text{final}} = \lambda_{\text{rate}}\mathcal{L}_{\text{rate}} + \lambda_{\text{dist}}\mathcal{L}_{\text{dist}} + \lambda_{\text{diff}}\mathcal{L}_{\text{diff}} + \lambda_{\text{sem}}\mathcal{L}_{\text{sem}}, \tag{6}$$

where $\lambda_{\text{rate}}$, $\lambda_{\text{dist}}$, $\lambda_{\text{diff}}$, and $\lambda_{\text{sem}}$ are scalar weights balancing each term.

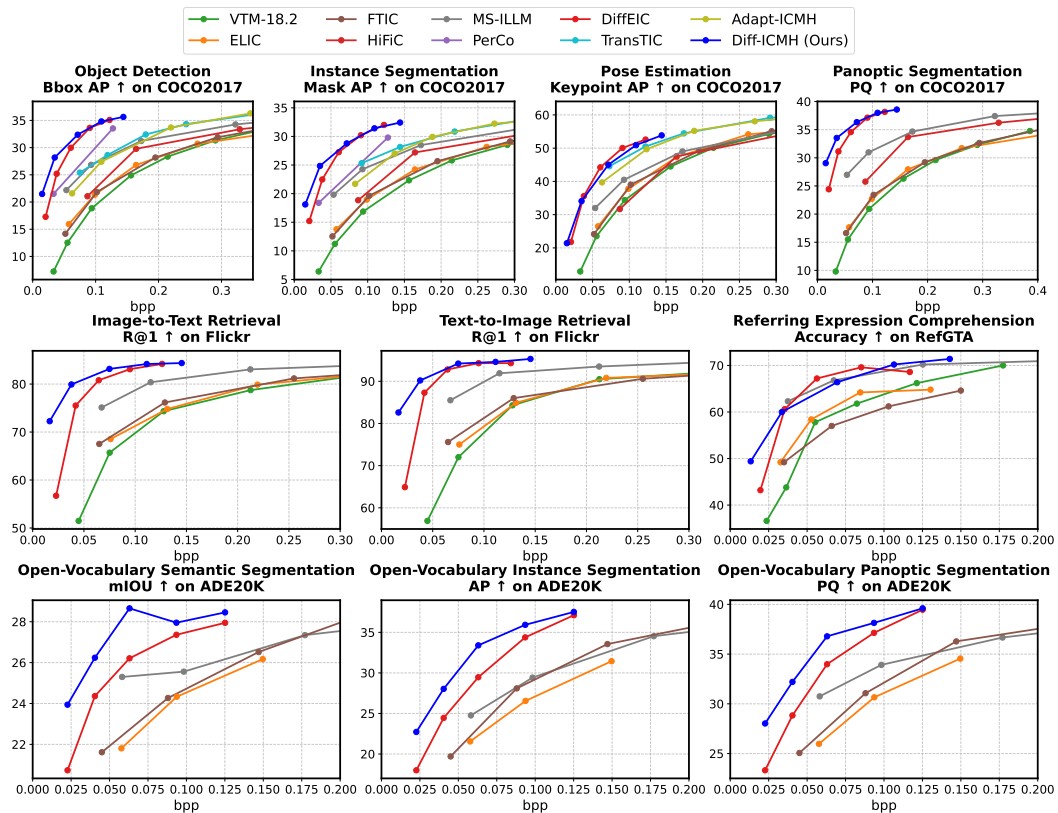

Figure 6: Performance comparison of Diff-ICMH with other methods across diverse intelligent tasks.

## 5 Experiments

### 5.1 Implementation details

**Dataset.** We use LSDIR [95] as our training dataset. During training, images from the dataset are randomly cropped to a size of $512 \times 512$.

**Model setup.** The experiments use Stable Diffusion 2.1 [1] as the pre-trained diffusion model. The control module is initialized in the same way as ControlNet [86], by copying parameters from the Stable Diffusion model for initialization, and initializing the weights of the Zero Convolution [86] to 0. During training, the pre-trained Stable Diffusion model and the tag extractor $\mathcal{E}_t$ remain frozen, while the encoder-decoder and control module parameters are learnable. For inference, we follow the standard DDPM process [96, 62] and start denoising from pure Gaussian noise.

**Hyper-parameters.** In equation (6), $\lambda_{\text{dist}}$, $\lambda_{\text{diff}}$, and $\lambda_{\text{sem}}$ are empirically set to 1, 1, and 2 respectively, while $\lambda_{\text{rate}}$ is set to $2, 4, 8, 16, 32$ to obtain codec models at different bit rates. We use the Adam optimizer [97] with $\beta_1$ and $\beta_2$ set to $0.9$ and $0.999$ respectively, and a training batch size of 16. Training is conducted in two stages. First, we train for 200K iterations with $\lambda_{\text{rate}} = 2$ and a learning rate of $1e-4$ to obtain a high bit rate model. Then, we fine-tune for another 200,000 iterations with a learning rate of $5e-5$ across all $\lambda_{\text{rate}}$ values to obtain models for all bit rate points. For Classifier-Free Guidance (CFG), text tags are dropped with a probability of $0.1$ during training. During inference, the CFG Scale is set to $5.0$. Inference uses DDIM sampling [98] with 50 steps.

### 5.2 Evaluation protocol

We conduct a comprehensive evaluation of Diff-ICMH across two key dimensions: performance on a diverse set of downstream intelligent tasks and the perceptual quality of image reconstruction.

---

[1]`https://github.com/Stability-AI/stablediffusion`

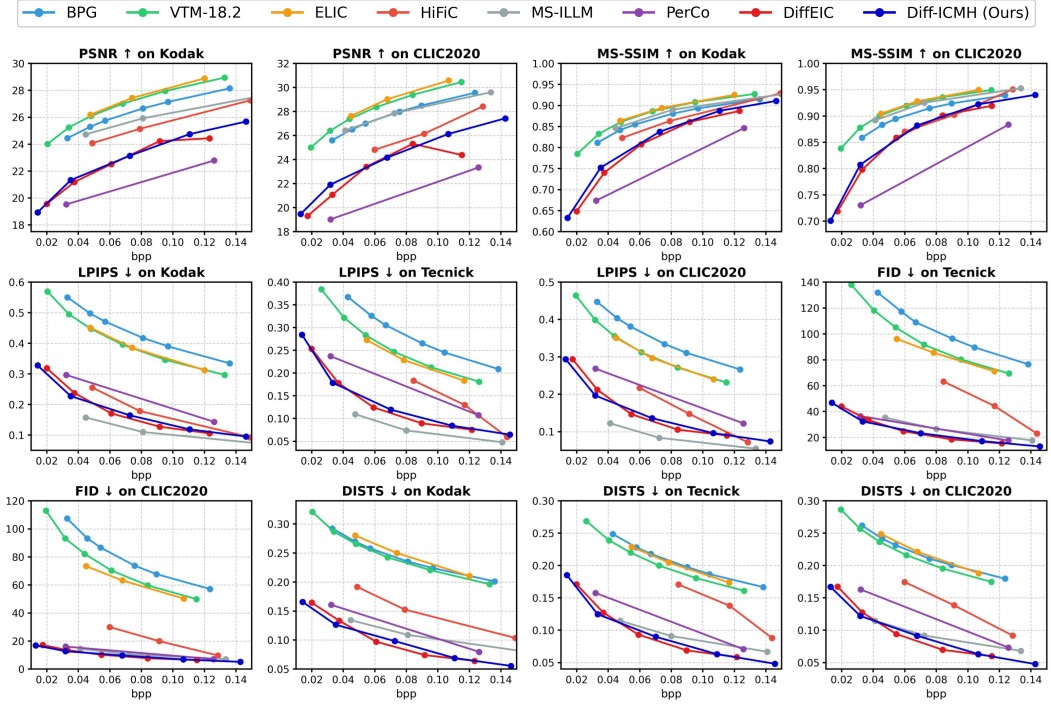

Figure 7: Comparison of Diff-ICMH and other methods on signal fidelity-oriented (PSNR↑, MS-SSIM↑) and perception-oriented (LPIPS↓, FID↓, DISTS↓) metrics.

**Intelligent tasks.** To evaluate the generalizability and effectiveness of our approach for machine vision, we conducted extensive experiments on a diverse range of intelligent tasks, encompassing a variety of task categories, model architectures, and backbone networks. These tasks includes traditional computer vision (e.g., detection and segmentation on COCO 2017 [99]), multimodal retrieval (Flickr30K [100]), and advanced Multimodal Large Language Model (MLLM) based understanding tasks (such as referring expression on RefGTA [101] and open-set segmentation on ADE20K [102]). Comprehensive details regarding these tasks are available in the Appendix.

**Perceptual quality of reconstruction.** Three public datasets are utilized: Kodak [103], Tecnick [104], and CLIC2020 [105]. We assess performance using metrics that measure signal fidelity (PSNR and MS-SSIM) as well as those that correlate with human perceptual quality (LPIPS, FID, and DISTS).

**Compared methods for intelligent task support.** We conduct comparative experiments with multiple high-performance codecs, including the powerful traditional image codec VVC [5](VTM-18.2), learned codecs ELIC [106] and FTIC [20], as well as codecs focusing on perceptual quality optimization, including HiFiC [71], PerCo [73, 107], MS-ILLM [72], DiffEIC [74], and task-specific optimized codecs TransTIC [44] and Adapter-ICMH [43] as comparative methods.

**Compared methods for perceptual quality.** We conduct comparisons with the traditional codec BPG, VVC [5] (VTM-18.2), the learned codec ELIC [106], and perceptual quality optimization codecs including HiFiC [71], PerCo [73, 107], MS-ILLM [72], and DiffEIC [74]. Numerical results of compared methods are obtained from the DiffEIC[2] repository.

### 5.3 Results

**Multiple intelligent task supporting.** Figure 6 illustrates Diff-ICMH's performance across diverse intelligent tasks on COCO, Flickr30K, RefGTA, and ADE20K datasets, compared against existing methods. Overall, Diff-ICMH consistently achieves state-of-the-art (SOTA) or highly competitive results. On COCO, it generally excels in object detection, instance, and panoptic segmentation, outperforming traditional, perception-oriented, and several task-specific codecs. For Flickr30K

---

[2]https://github.com/huai-chang/DiffEIC

cross-modal retrieval and ADE20K open-vocabulary segmentation tasks, Diff-ICMH demonstrates significant advantages, particularly at very low bitrates (e.g., 0.01-0.05 bpp on Flickr, 0.02-0.1 bpp on ADE20K), underscoring its effective semantic preservation capabilities attributed to our SC loss and TGM. This highlights strong adaptability to both traditional vision and advanced MLLM-based understanding tasks. While generally leading, performance is comparable to some methods like DiffEIC in specific scenarios, such as COCO pose estimation or RefGTA referring expression, potentially due to inherent VAE latent space limitations for fine details or domain shift from synthetic data, respectively. Despite these minor trade-offs, the extensive evaluations confirm Diff-ICMH's broad generalizability and excellent overall rate-distortion performance across a multitude of tasks without requiring any task-specific fine-tuning, emphasizing its practical value.

**Perception-oriented reconstruction.** Figure 7 presents the compared results of reconstruction quality. Consistent with its generative nature, Diff-ICMH's PSNR/MS-SSIM scores trail behind traditional fidelity-optimized codecs, a common characteristic when prioritizing perceptual realism. Besides, it is comparable to DiffEIC and outperforms PerCo in these fidelity metrics. Conversely, Diff-ICMH demonstrates substantial advantages in perceptual quality. It significantly surpasses fidelity-focused codecs (e.g., BPG, VTM-18.2) and other perception-oriented methods (e.g., HiFiC, PerCo) across LPIPS, FID, and DISTS. Notably, Diff-ICMH achieves SOTA performance in FID and DISTS scores among all compared methods, exhibiting particular strength at extremely low bitrates. These perceptual gains, especially at low BPP, are attributed to the effectiveness of our SC loss and TGM in preserving semantic integrity and guiding realistic reconstruction.

**Additional results.** For brevity, *visualizations*, *feature difference analysis*, and *computational complexity analysis* are provided in Section B.2, B.3, B.4.

## 5.4 Ablation study

We conducted ablation studies to validate the effectiveness of the proposed Semantic Consistency loss (SC loss) and Tag Guidance Module (TGM), along with an analysis of the SC loss setup. Models are trained on the LSDIR dataset and evaluated using object detection on COCO 2017. More details are provided in the Appendix.

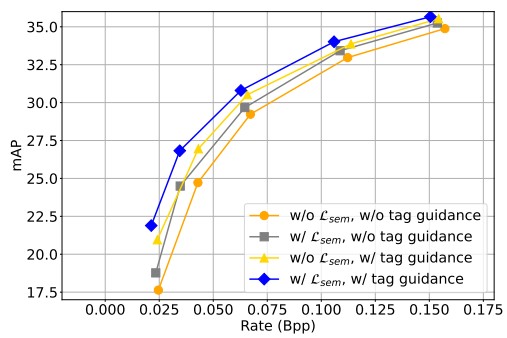

Figure 9: Ablation of Semantic Consistency loss and Tag Guidance Module.

**Effectiveness of SC loss and TGM.** Figure 9 presents ablation results for the SC loss and TGM. Both modules significantly improve the task performance, with their combination (blue curve) yielding the best performance (e.g., around 4 mAP gain over the baseline at approximately 0.025 Bpp). This confirms their individual contributions and synergistic effect. These results validate that SC loss maintains semantic similarity while TGM activates generative priors, complementarily boosting performance.

**Setup of SC loss.** Figure 8 details ablations for SC loss hyperparameters.

- **Weight $\lambda_{\text{sem}}$.** As shown in Figure 8 (a), $\lambda_{\text{sem}} = 2.0$ achieves the optimal balance, effectively preserving semantic information while maintaining low bitrate.

- **Position of SC loss application.** Applying SC loss to deeper features within the U-Net yields better rate-distortion performance (Figure 8 (b)). The middle block is found to be optimal, as features at this depth better capture abstract semantic content compared to shallower layers.

- **Input noise level for SC loss.** We compared using clean latent features $(\mathbf{z}, \hat{\mathbf{z}})$ versus noised versions $(\mathbf{z}_t, \hat{\mathbf{z}}_t)$ as inputs to the semantic feature extractor $f(\cdot)$ for the SC loss. Figure 8 (c) demonstrates that noise-free inputs $(t = 0)$ result in the optimal rate-distortion performance. This suggests that while the diffusion model is trained with noise, its capability for semantic feature extraction (for our SC loss) is maximized with clean signals. Notably, all tested noise levels still improved performance over the baseline, confirming the robustness of the SC loss.

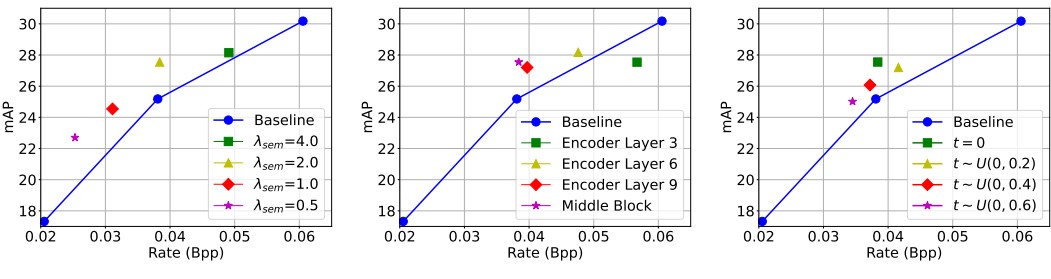

(a) Weight $\lambda_{\text{sem}}$ of SC loss.  (b) Position of SC loss application.  (c) Noise level $t$ for SC loss inputs.

Figure 8: **Ablation of hyper-parameters of SC loss.** "Baseline" indicates not including SC loss.

# 6 Limitation

The primary limitation of Diff-ICMH is computational complexity, as discussed in Section B.4. Diff-ICMH requires relatively significant decoding time due to the iterative nature of the diffusion denoising process, where each step necessitates a complete forward pass through the neural network. However, we emphasize that the primary focus of this paper is to demonstrate the generalizability and significant potential of Diff-ICMH for harmonizing diverse intelligent tasks with human perceptual quality. Therefore, computational complexity analysis is not the central emphasis of our current work. In future work, incorporating efficient sampling methods [108–110] and distillation techniques [111–113] to reduce sampling steps would make Diff-ICMH more practical for real-world applications.

# 7 Conclusion

In this paper, we introduced Diff-ICMH, a verstile generative image codec designed to effectively serve both intelligent tasks and human visual perception. The core design philosophy of Diff-ICMH departs from isolated optimization by revealing and leveraging fundamental commonalities between these two objectives: Preserving semantic information integrity is paramount for both machine analysis and human understanding. Concurrently, enhanced perceptual quality, which is built upon this semantic foundation, not only improves the visual experience but also benefits machine feature extraction. Diff-ICMH embodies this by integrating a diffusion model-based generative framework with a novel Semantic Consistency loss and an efficient Tag Guidance Module. Comprehensive evaluations demonstrate the effectiveness and generalization of our proposed method, which efficiently supports diverse tasks with a single codec and no task-specific adaptation, alongside excellent visual quality. Our work thus presents a promising pathway towards truly versatile image compression.

**Acknowledgements**  This work was supported in part by NSFC under Grant 62371434, 623B2098, and 62021001, the Postdoctoral Fellowship Program of CPSF under Grant Number GZC20252293, and the China Postdoctoral Science Foundation-Anhui Joint Support Program under Grant Number 2024T017AH. This work was also funded by Anhui Postdoctoral Scientific Research Program Foundation (No.2025A1015).

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

# Appendix

## Contents

## A   Implementation Details

### A.1   Architecture of the control module

The control module described in Section 4.1 is modified from original ControlNet [86]. Key modifications include: (i) Replacing ControlNet's initial three stride-2 convolutions with a single stride-1 convolution to establish dimensional compatibility with $\hat{\mathbf{z}}$. (ii) Implementing bilateral feature injection through zero-convolutions from both encoder and decoder pathways of the diffusion model (unlike ControlNet's decoder-only approach) to establish more precise control mechanisms from $\hat{\mathbf{z}}$. Additionally, the extracted word-level tags $\mathbf{c}$ are strategically injected as text prompts into both the control module and the diffusion model to effectively activate relevant generative priors.

### A.2   Implementation in ablation studies

**Basic setup.** As described in Section 5.4, we designed ablation experiments to verify the effectiveness of the SC loss and tag guidance module, along with the hyper-parameter configuration of SC loss. In the experimental setup, training was conducted on the LSDIR dataset with a batch size of 8, a total of $80,000$ iterations, and a learning rate of $1e-5$. The experimental results were rigorously evaluated on the COCO 2017 object detection benchmark dataset.

**Semantic Consisitency loss with noisy input.** In Section 4.2, for SC loss calculation, clean latent features $\mathbf{z}$ and $\hat{\mathbf{z}}$ are utilized as inputs to the diffusion model. Recognizing that diffusion models are typically trained on noisy signals, we conducted a series of comparative experiments (illustrated in Figure 10) to investigate the implications of using noisy latent features $\mathbf{z}_t$ and $\hat{\mathbf{z}}_t$ as inputs, corresponding to results in Section 5.4, "Input noise level for SC loss". Specifically, in this variant, the same sampled noise is first added to the latent variable $\mathbf{z}$ and the reconstructed latent variable $\hat{\mathbf{z}}$ using the same timestep $t$ sampled from $U(0, t_{\max})$, where $t_{\max}$ is a predefined threshold used to control the maximum noise intensity. A value of $t=1$ corresponds to the complete 1000-step noising process. These are then separately fed into the trained diffusion model for feature mapping to obtain the corresponding semantic features, which are subsequently aligned.

**Bits of Tag Guidance.** We employ RAM++ from Recognize Anything [93], which has a maximum default vocabulary of 4585 tags. For simplicity, we utilize fixed-length encoding without entropy coding, requiring 13 bits per tag ID to accommodate a maximum of 8192 tags. Based on our analysis of 500 randomly sampled COCO images, RAM++ predicts an average of 8.7 tags per image under its

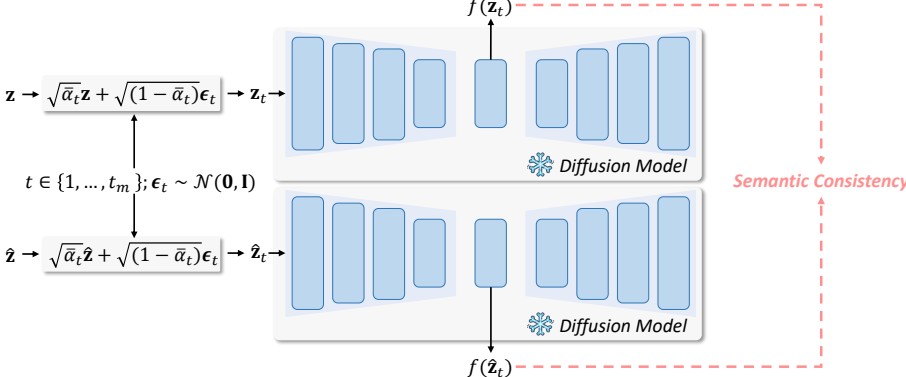

Figure 10: **Illustration of the variant of SC loss calculation.** Noisy latent variable $\mathbf{z}_t$ and $\hat{\mathbf{z}}_t$ are utilized as input into the pre-trained diffusion model.

default settings. Consequently, the average bit overhead for tag guidance amounts to $13 \times 8.7 = 113.1$ bits per image.

### A.3 Hardware device

All training and inference experiments are conducted on 4 NVIDIA A100 Tensor Core GPUs.

## B Experiments

### B.1 Details of evaluation protocol

We conduct a comprehensive evaluation from two dimensions: the performance on various intelligent tasks and the perceptual quality of image reconstruction.

**Intelligent tasks.** As mentioned in Section 5.2, the efficacy and generalization capabilities of our proposed method are rigorously evaluated across a diverse spectrum of intelligent tasks spanning different domains, datasets, task-specific models, and vision backbones. To the best of our knowledge, this is the first comprehensive study to present such extensive experimental validation in the context of machine-oriented image coding. We believe this thorough empirical analysis not only substantiates our contributions but also establishes new benchmarks that may accelerate advancement in this emerging field.

The specific information about datasets and task models is shown in Table 1, where datasets include COCO 2017 [99], Flickr30K[3], RefGTA [101], and ADE20K[4]. The intelligent tasks cover three major categories: traditional perception-based computer vision tasks, multimodal retrieval tasks, and multimodal understanding tasks based on large language models. For task models, traditional perception tasks are evaluated through a series of models in the Detectron2 toolkit; multimodal retrieval tasks are evaluated through the BEiT-3 model; and for multimodal understanding capabilities, we specifically selected two multimodal large language models (MLLMs) based on different backbone networks, processing different data domains, and executing tasks of varying granularities.

This comprehensive experimental framework aims to thoroughly verify the generalizability and superior performance of the Diff-ICMH approach across different data types, datasets, intelligent tasks, and task models. For evaluation metrics, we employ standard measures appropriate to each task: mAP (mean Average Precision) of bounding boxes for object detection, mAP of segmentation masks for instance segmentation, AP of keypoints for pose estimation, PQ (Panoptic Quality) for panoptic segmentation, Recall@1 for multimodal retrieval tasks, and accuracy for referring expression comprehension. The open-set pixel-domain understanding tasks based on MLLMs are evaluated using mIoU for semantic segmentation, mAP for instance segmentation, and PQ for panoptic segmentation.

---

[3]https://hockenmaier.cs.illinois.edu/DenotationGraph/
[4]https://ade20k.csail.mit.edu/

Table 1: Overview of experimental datasets, tasks, and models.

| Dataset | Task Type | Specific Task | Task Model | Backbone |
|---|---|---|---|---|
| COCO 2017 | Traditional Perception | Object Detection
Instance Segmentation
Pose Estimation
Panoptic Segmentation | Faster R-CNN
Mask R-CNN
Keypoint R-CNN
Panoptic-FPN | R50-FPN [114] |
| Flickr30K | Multi-Modal Retrieval | Image-Text Retrieval
Text-Image Retrieval | BEiT-3 [115] | MW-Transformer [116] |
| RefGTA
ADE20K | MLLM Understanding | Referring Expression
Open-Set Segmentation | Qwen2.5-VL[117]
Ospray [118] | WA-ViT [117]
ConvNext [119] |

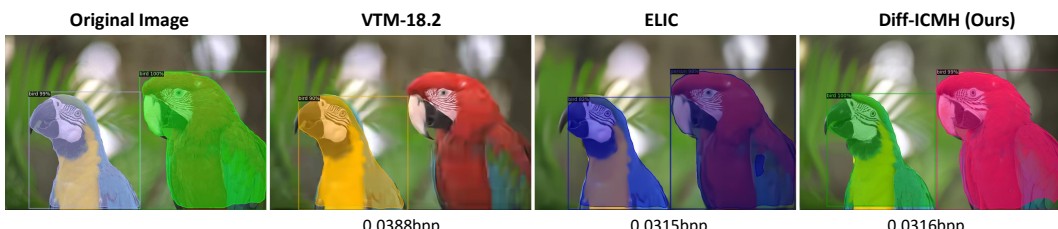

Figure 11: Visualized results on instance segmentation.

**Image reconstruction.** This paper uses three public datasets: Kodak [103], Tecnick [104], and CLIC2020 [105]. During the experiments, images from Tecnick and CLIC2020 datasets are rescaled to 768 pixels on the short side, and samples of size $768 \times 768$ are cropped from the center of the images for evaluation, following previous method [74]. Evaluation metrics include PSNR and MS-SSIM for measuring signal fidelity, as well as LPIPS, FID, and DISTS for evaluating perceptual quality.

## B.2 Visualization results

**Task supporting.** Figure 11 presents visualization results comparing our method against VTM-18.2 [5], ELIC [106], and FTIC [20] on instance segmentation tasks. The comparison reveals that VTM-18.2, despite operating at its highest bit rate, fails to correctly identify the two distinct objects within the image. Similarly, ELIC erroneously classifies the "bird" object on the right as a "person," demonstrating significant recognition inaccuracy. In contrast, only the reconstruction output generated by our Diff-ICMH method successfully preserves the critical semantic information necessary for accurate object identification, thereby effectively supporting the completion of this intelligent task with precision.

**Perception-oriented reconstruction.** Figure 12 shows the visual comparison of reconstruction quality between Diff-ICMH and other compression methods. From the comparison, it can be observed that methods optimized for signal fidelity (VTM-18.2, ELIC, FTIC) produce reconstructed results with obvious blurring, significantly reducing visual quality. Although MS-ILLM shows some improvement in texture clarity, its reconstructed textures lack authenticity and are accompanied by obvious noise interference. In contrast, our proposed Diff-ICMH demonstrates the best visual realism in reconstruction results while maintaining clear boundary contour features.

## B.3 Feature difference analysis

To further evaluate Diff-ICMH's performance advantages in semantic information protection and intelligent task support, we designed comparative experiments based on feature difference analysis. Specifically, we calculated the differences between feature representations of reconstructed images from various compression methods and those of original images at different layers of a ResNet50 feature extractor (using 1 minus cosine similarity as the metric, where lower values represent higher feature similarity), and plotted the curves of these differences across network depths.

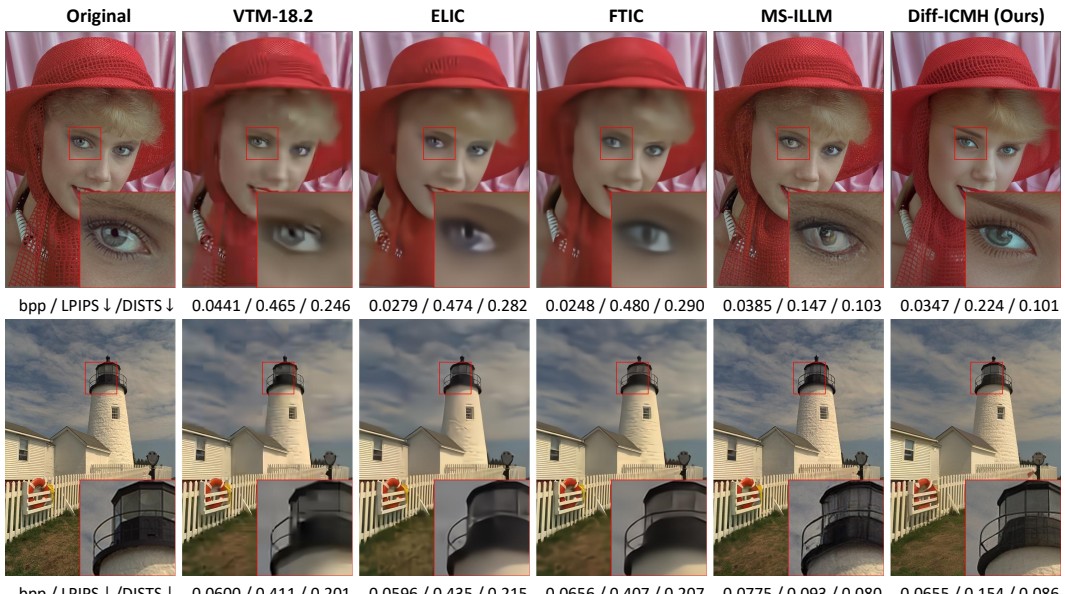

| | Original | VTM-18.2 | ELIC | FTIC | MS-ILLM | Diff-ICMH (Ours) |
|---|---|---|---|---|---|---|
| bpp / LPIPS↓ /DISTS↓ | | 0.0441 / 0.465 / 0.246 | 0.0279 / 0.474 / 0.282 | 0.0248 / 0.480 / 0.290 | 0.0385 / 0.147 / 0.103 | 0.0347 / 0.224 / 0.101 |

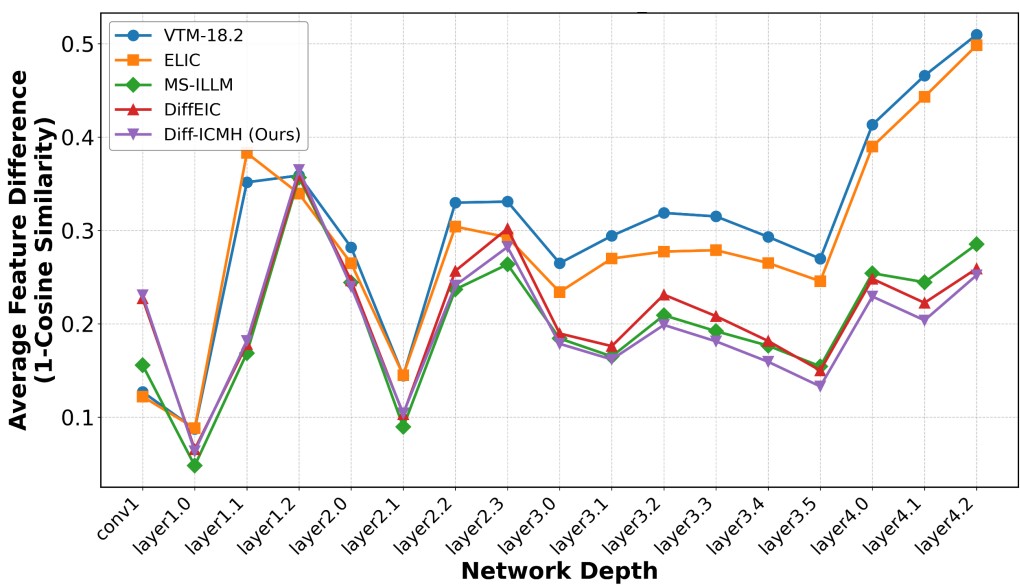

bpp / LPIPS↓ /DISTS↓     0.0600 / 0.411 / 0.201     0.0596 / 0.435 / 0.215     0.0656 / 0.407 / 0.207     0.0775 / 0.093 / 0.080     0.0655 / 0.154 / 0.086

Figure 12: Reconstruction of Diff-ICMH and other compression methods.

Figure 13: The evolution curves showing feature differences between reconstructed images from Diff-ICMH and other compression methods compared to original images across various layers of the ResNet50 feature extractor. Lower values represent higher feature similarity.

The experiments are conducted on the Kodak dataset, analyzing the average results of all images. As shown in Figure 13, the experimental results exhibit significant hierarchical characteristics: in shallow network layers, generative compression methods (MS-ILLM, DiffEIC, Diff-ICMH) show relatively higher difference values, while methods optimized for signal fidelity (VTM-18.2, ELIC) maintain minimal differences. However, as the network hierarchy and semantic level deepen, this pattern changes significantly—the latter methods' difference values increase rapidly, reaching a peak of approximately 0.5 at the deepest layer, far exceeding generative compression methods.

Among generative compression methods, Diff-ICMH's advantage is demonstrated in its performance from the middle network layers (layer3.0) to the deepest layers (layer4.2), consistently maintaining the

Table 2: Encoding and decoding time on Kodak dataset. (Second)

| Method | NFE | Encoding Time | Decoding Time | Hardware |
|---|---|---|---|---|
| VVC | - | 13.862 | 0.066 | 13th Core i9-13900K |
| ELIC | - | 0.056 | 0.081 | RTX4090 |
| HiFiC | - | 0.038 | 0.059 | RTX4090 |
| MS-ILLM | - | 0.038 | 0.059 | RTX4090 |
| PerCo | 5 | 0.080 | 0.665 | A100 |
| PerCo | 20 | 0.080 | 2.551 | A100 |
| DiffEIC | 20 | 0.128 | 1.964 | RTX4090 |
| DiffEIC | 50 | 0.128 | 4.574 | RTX4090 |
| Diff-ICMH | 20 | 0.232 | 5.456 | A100 |
| Diff-ICMH | 50 | 0.232 | 13.14 | A100 |

lowest feature difference values in this range. This result confirms the superiority of our approach in semantic information protection. More importantly, considering that the feature extractor (ResNet50) used in the experiment and the feature mapping (Stable Diffusion) used in the SC loss belong to different network architectures and different pre-training purpose, this performance advantage also highlights the excellent model generalization capability of our method.

## B.4 Complexity analysis

Table 2 illustrates the encoding and decoding time of different methods. The data reveals that diffusion-based methods (such as DiffEIC and Diff-ICMH) perform well in terms of encoding speed, showing significant advantages compared to traditional VVC methods (13.862 seconds), with DiffEIC taking 0.128 seconds and Diff-ICMH taking 0.232 seconds.

However, there is room for improvement in our method's decoding time. Particularly, when the number of denoising steps increases, the decoding time increases significantly. For instance, Diff-ICMH requires 5.456 seconds and 13.14 seconds for decoding at 20 steps and 50 steps, respectively. The decoding time is considerably longer than traditional methods like VVC (0.066 seconds) and GAN-based methods such as HiFiC (0.038 seconds) and MS-ILLM (0.059 seconds).

The main reason for this phenomenon is that the progressive denoising process of diffusion models is inherently an iterative computational process, with each step requiring a complete network forward pass. When the number of steps increases, the computational cost increases linearly. Optimizing encoding and decoding speeds will be an important direction for future work: (i) Investigating more efficient sampling strategies, such as implementing samplers with less steps [108, 109]; (ii) exploring knowledge distillation techniques to distill multi-step denoising networks into lighter single-step or few-step networks [111]; (iii) and optimizing network structures to reduce the computational complexity of each denoising step.

While diffusion-based image methods currently face computational efficiency challenges, this work primarily serves to demonstrate the fundamental potential of this paradigm. We anticipate that ongoing advancements in sampling algorithm efficiency and targeted engineering optimizations will substantially reduce encoding and decoding latency, thereby enhancing the practical utility of diffusion models in this research field.

## B.5 Ablation study of distortion loss calculation space

In Equation (4) of the main text, the distortion loss $\mathcal{L}_{\text{dist}}$ are calculated in the VAE latent space:

$$\mathcal{L}_{\text{dist}} = \|\mathbf{z} - \hat{\mathbf{z}}\|_2^2 = \|\mathcal{E}_{\text{VAE}}(\mathbf{x}) - \mathcal{D}_c(\hat{\mathbf{y}})\|_2^2, \quad (7)$$

Here we conduct ablation study of calculating loss in the pixel space:

$$\mathcal{L}_{\text{dist}} = \|\mathbf{x} - \hat{\mathbf{x}}\|_2^2 = \|\mathbf{x} - \mathcal{D}_c'(\hat{\mathbf{y}})\|_2^2, \quad (8)$$

where $\mathcal{D}_c'$ maintains the same architectural foundation as $\mathcal{D}_c$ but incorporates additional upsampling blocks

21

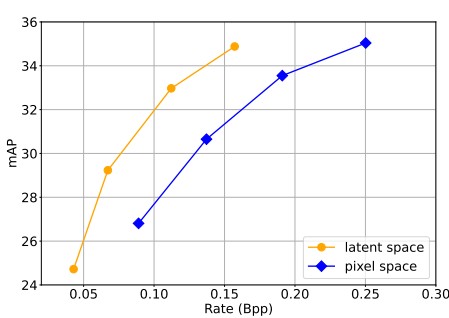

Figure 14: Ablation of distortion loss calculation space.

to reconstruct signals at pixel-level resolution. The re-
constructed pixel-space output $\hat{x}$ is subsequently fed
into the control module for generative reconstruction.

Figure 14 demonstrates the substantial performance
advantage of calculating distortion in the latent space rather than pixel space. This finding confirms
that the VAE latent space provides a more compact and perceptually meaningful representation that
effectively filters semantically irrelevant information from the pixel domain. Through optimizing
fidelity in the latent space, the compressed bitstream effectively prioritizes semantically salient infor-
mation, thereby achieving superior rate-distortion performance across various downstream machine
vision applications.

