# OpenReview forum: "Diff-ICMH: Harmonizing Machine and Human Vision in Image Compression with Generative Prior"
_NeurIPS.cc/2025/Conference — NeurIPS 2025 poster_

### Official Review · Reviewer_fiP4 · 2025-06-07

**Clarity:** 2
**Significance:** 2
**Originality:** 2
**Rating:** 4
**Confidence:** 4

**Summary:**

The authors propose that image compression meant for downstream tasks should be built accordingly to emphasize more semantic fidelity. They propose Diff-ICMH, a generative image compression method based on pretrained diffusion models that tries to do just that, while also optimizing for perceptual quality. Specifically, they train a new encoder-decoder to compress the latent representation of a given image (incorporating a semantic consistency loss), and on the client side decompress the image by guiding the diffusion model with image tags (also transmitted) and by using controlNet to condition the model on the decompressed latent vector. They show strong performance on downstream tasks.

**Questions:**

- L251: The paper suggests that tag-based conditioning incurs minimal overhead, but explicit numbers would help quantify this claim. 100 bits for a 40K words dictionary would be less than 10 words? The authors mention they set a limit, this should be reported.
- The authors use Kodak images, but considering their size (768×512), how well do they align with the 768x768 SD 2.1?
- L183: From my knowledge and experience, the VAE latent space isn't very smooth (e.g. [3]). It baffles me that optimizing similarity in this latent space helps enforce actual similarity. How do you explain that? Also, how does this affects the robustness of the downstream tasks results?

[3] Norm-guided latent space exploration for text-to-image generation. D Samuel, R Ben-Ari, N Darshan, H Maron, G Chechik - Advances in Neural Information Processing Systems, 2023.

**Ethical Concerns:**

["NO or VERY MINOR ethics concerns only"]

**Final Justification:**

The authors answered most of my concerns, importantly:
- clarifying figures 7 & 5
-  Will add ablations over different tasks
- fix details such as referring to PerCo (SD), and consistent step counts, referring to DiffEIC metrics, speed on same devices
- promised to add more visualization and move at least one to the main text
- limitations will be added to the paper concretely.

However, the novelty is still somewhat weak in my eyes. Fig 1 is also still a bit redundant in my opinion, and the authors should at least make it more compact.

**Limitations:**

No, the authors mention in the checklist they do in the appendix, but this is very briefly touched upon only in a complexity analysis. No mention at all about the different training needed for different bit-rates.

**Quality:**

3

**Strengths And Weaknesses:**

## Strengths
- Using tags instead of automatically generated sentences (e.g. by using BLIP) is a good idea.
- The proposed method exhibits strong performance across a diverse range of downstream tasks, demonstrating its adaptability and effectiveness.
- I believe using pre-trained diffusion models is a good direction for compression algorithms, leveraging their generative capacity to enhance visual quality.
- Faire reporting on PSNR comparing non-latent based methods (showing that using a VAE induces limitations).

## Weaknesses
- The novelty is somewhat unclear to me. The claim that combining perceptual optimization and Image Compression for Machines (ICM) is novel is somewhat questionable. The concept of using a single codec for multiple downstream tasks has been previously explored (e.g., DDCM [1]). Furthermore, traditional codec-based methods arguably serve similar roles and should not be excluded from such categorization.
- Motivation: In L110 the authors argue that traditional image compression methods harm semantic integrity, but I see no evidence that they perform poorly on that front. On the contrary, perceptually oriented methods inherently maintain semantic accuracy, making this assertion debatable.
- **Clarity Issues and Structural Organization:**
  - Figure 1 is unclear to me, conveying very limited information (and the caption does not explain as well). I don't see why it should occupy so much space in the main paper. Instead, the authors should use that space for the image examples they lack.
  - Section 4.1 – It's unclear if the generative process follows the standard diffusion process (does $z_t$ starts from random noise here)?
  - It's not explicitly stated what information is transmitted and compressed, and the reader must infer it consists of the compressed latent space representation alongside the tag bits. This should be clearly written.
   - Figure 7 lacks organization. Metrics are arranged seemingly at random, making it difficult to follow ((E.g., Kodak PSNR is at [0,0], its LPIPS is in [1,0], it's MS-SSIM is [0,2], and DISTS in [1,2]). Not all metrics mentioned in the caption are shown for each DB, e.g. Kodak lacks FID. If these are omitted due to the number of images in the DB, patches should be used instead (this is quite commonplace, see e.g. [1])
  - Figure 5: the arrow comes out from the bottleneck of the diffusion model, which suggests the semantic space you consider is the h-space of the diffusion model. The text however implies it is the VAE latent space. Please clarify. Additionally, Eq 1 seems to suggest this is not happening per time step, which confuses me about when the loss is performed.
  - Ablations in Section 5.4: The mAP metric lacks context—was it calculated on a specific task or averaged across multiple tasks? L276 mentions better rate-distortion performance, but the evidence supporting this claim is unclear.
- The paper includes multiple comparisons to PerCo. However, their internal model was not released. Did the authors train their own PerCo version, or used the open sourced version, "Perco (SD)"? If the latter, the corresponding paper [2] should be cited.
- The DDIM step count differs between methods—Diff-ICMH utilizes 50 steps, whereas PerCo is restricted to 20 steps. PerCo’s perceptual quality may suffer from this shorter inference process. A fairer comparison using equivalent step counts would be preferable.
- The ablations are lacking with regards to the necessity of TGM. Recent methods seem to not need text guidance for similar bitrates ([1]).
- Visualizations: The decision to limit visual examples to two and relegate perceptual comparisons to the appendix is problematic, to say the least. Given the claim of perceptual superiority, a more thorough presentation in the main paper is warranted, along with additional examples in the appendix. Furthermore, these perceptual comparisons (Fig 12) do not compare to SOTA diffusion based methods such as DiffEIC, PerCo or DDCM, raising concerns regarding the fairness of comparisons to similar generative priors. Also, in the first image the proposed method's BPP is one of the highest in the figure, leading to unfair comparison, for a fair comparison the proposed method should have the same BPP roughly or lower. Finally, the zoomed region in Figure 12 would be better positioned as an additional row rather than obscuring portions of the original image.
- The results in Figure 7 appear identical to those reported in DiffEIC. If data reuse occurred, this should be explicitly acknowledged and cited. Is that the case? If so, the authors should verify their evaluation protocol to ensure the reuse did not affect the validity of the reported performance.
- The paper lacks discussion on trained, partially trained, and zero-shot compression methods, which would provide a more balanced perspective on result comparisons (L251)
- Table 2 reports speed comparisons across different hardware settings, which does not make sense.
- This method requires training (or fine-tuning) for every new desired bit-rate.

[1] Compressed Image Generation with Denoising Diffusion Codebook Models. G Ohayon, H Manor, T Michaeli, M Elad - arXiv preprint arXiv:2502.01189, 2025

[2] PerCo (SD): Open Perceptual Compression. N Körber, Workshop on Machine Learning and Compression, NeurIPS 2024.

---

> ### Author Rebuttal · Authors · 2025-07-28
>
> **Rebuttal to Weaknesses**
> - About "Novelty". To our knowledge, we are the first to explore a single codec with a single bitstream that adapts to different intelligent tasks and human perception. By combining generative priors, we designed SC Loss and TGM to achieve this goal. Besides, we have carefully reviewed the DDCM paper. This work primarily focuses on compression algorithms for perceptual quality and does not include any exploration on intelligent tasks. We will cite this paper and provide distinction and discussion. Finally, we show the pipeline of adapting traditional codecs to different tasks in the first diagram of Figure 1 and discuss it in L30-L35. Although the traditional codec itself is general, the bitstream is task-specific. This is because they often optimize for specific tasks, such as some RoI bit allocation methods, whereas Diff-ICMH does not require any additional design or separate bitstreams.
> - About "Motivation". We sincerely apologize for any misunderstanding. We must emphasize that in the paper, we do not make any claim that traditional image compression methods "harm" semantic fidelity; they simply do not focus on semantic fidelity. Nor do we claim that perceptual-oriented methods can inherently better preserve semantic accuracy. In the motivation section, our claim is that two key points for intelligent tasks are semantic fidelity and perceptual realism. The former directly relates to whether tasks can be completed normally, while the latter can cause error accumulation and further semantic distortion due to training-inference mismatch. Traditional signal fidelity-oriented image compression methods only optimize for signal fidelity itself and do not focus on semantic fidelity and perceptual realism. These methods include both traditional codecs and learning-based codecs. Perceptual-oriented methods can achieve better performance compared to pure signal fidelity-oriented image compression methods because they better ensure perceptual realism. Diff-ICMH aims to simultaneously address both semantic fidelity and perceptual realism mentioned above. We will further emphasize this statement in the paper.
> - About "Clarity Issues and Structural Organization".
>   - "Necessity of Figure 1". We must emphasize that Figure 1 is a very important illustration. As we introduce in L30-L40, previous codecs for intelligent tasks either require multiple codecs to achieve this purpose, or different intelligent tasks need different bitstreams. Therefore, the original starting point of this work, as shown in the last diagram of Figure 1, is to use a unified codec with a generalized bitstream to adapt to different intelligent tasks as well as human vision.
>   - "generative process follows the standard diffusion process?" Yes. We will describe this more clearly in the paper.
>   - "not explicitly stated what information is transmitted". As illustrated in Figure 4, the transmitted bitstream consists of the compressed latent space representation and tag bits. We will accurately describe this linguistically in Section 4.1.
>   - "Figure 7 lacks organization." Thank you for your suggestion. We will reorganize the arrangement of these figures to present the results more clearly.
>   - "where the SC loss is conducted". Thank you very much for your careful review. The semantic space refers to the feature space of the diffusion model, as illustrated in Figure 5. We will revise the wording for greater clarity in the paper.
>   - "Ablations in Section 5.4". Regarding the experimental setup for ablation studies. As stated in L262, the experimental details for the ablation study correspond to section A.2 in the appendix: "The experimental results were rigorously evaluated on the COCO 2017 object detection benchmark dataset." Regarding the results, as illustrated in Figure 8 (b), results of the middle block achieves the best RD performance (in the Rate-Distortion performance graph, closer to the upper left indicates better performance), which demonstrates our claim.
> - "Did the authors train their own PerCo version, or used the open sourced version?" We adopted Perco(SD): https://github.com/Nikolai10/PerCo. We will cite and acknowledge it.
> - "The DDIM step count differs between methods". Experiments with steps=20 of Diff-ICMH and PerCo are shown below. It can be observed that our method also significantly outperforms PerCo with the same denoising steps.
> | Method    | bpp    | mAP   |
> |-----------|--------|-------|
> | Diff-ICMH | 0.1092 | 34.68 |
> |   | 0.0350 | 27.88 |
> | PerCo  | 0.1274 | 33.53 |
> |  | 0.0332 | 21.50 |
>
> - "The ablations are lacking with regards to the necessity of TGM". We sincerely apologize for any misunderstanding. We must emphasize that we did not omit the ablation study for TGM. Figure 9 and L263-L272 correspond to the ablation study experimental results and analysis for TGM and SC Loss. We will change "tag guidance" to "TGM" in Figure 9 to avoid similar misunderstandings.
> - "Visualizations"
>   - "A more thorough presentation in the main paper ... along with additional examples in the appendix". Excellent suggestion. We will reorganize to include key perceptual results in the main paper and expand the Appendix with additional examples.
>   - "perceptual comparisons (Fig 12) do not compare to SOTA diffusion based methods such as DiffEIC, PerCo or DDCM". Thanks for your suggestion. We will add comparison results with those SOTA diffusion based methods.
>   - "in the first image the proposed method's BPP is one of the highest in the figure". Thank you for your insightful feedback. We will adjust the bitrates in our comparisons for improved clarity.
>   - "zoomed region in Figure 12 ... additional row". We appreciate this constructive suggestion. We will add additional rows of zoomed-in images for each comparison group to enable more precise visual analysis.
> - "data reuse". Yes, the experimental results in Figure 7 are consistent with DiffEIC. We will explicitly acknowledge and cite it.
> - "lacks discussion on trained, partially trained, and zero-shot compression methods". We seek clarification on the comment about "trained, partially trained, and zero-shot compression methods." Standard neural codec evaluation requires fully trained models for meaningful comparison. Partially trained or zero-shot approaches would significantly degrade performance and provide uninformative comparisons.
> - "speed comparisons across different hardware settings". Yes, the evaluation is conducted across different hardware configurations. However, we emphasize that the primary focus of this paper is to demonstrate the generalizability and significant potential of Diff-ICMH for diverse intelligent tasks and human perceptual quality assessment. Therefore, computational complexity analysis was not the central emphasis of our current work.
> - "requires training for every new desired bit-rate." Yes, each rate point currently requires a separate trained model. However, most methods not specially designed for variable-rate compression also use one model per bit-rate, which is the standard validation approach for novel neural compression techniques. Our primary contribution is demonstrating Diff-ICMH's generalizability across diverse intelligent tasks and human perceptual quality assessment. While codecs equipped with variable-rate adjustment can achieve multiple rates with a single model, such engineering implementations are beyond our current research scope.
>
> **Answers to Questions**
> - "details of explicit numbers of tag conditions". Thanks for your constructive suggestion. We are using the RAM++ version from Recognize Anything (https://github.com/xinyu1205/recognize-anything). Its maximum default vocabulary is 4585, and we do not perform any entropy coding but use fixed-length encoding, so each ID requires 13 bits for compression (max as 8192). Additionally, we counted that each image on COCO predicts 8.7 tags on average under RAM++'s default setting, so the final average bit count is 13×8.7=113.1 bits. We will supplement these specific details in the article and supplementary materials. Thank you again for your suggestion.
> - "resolution mismatch". Thank you for this insightful observation. You're correct that stable diffusion models trained on fixed resolutions can suffer from training-inference mismatch when generating different resolutions, often leading to quality degradation and structural collapse. However, our approach incorporates ControlNet, which provides additional control signals that effectively address these structural issues. This enables robust adaptation across different resolutions. As demonstrated in the original ControlNet paper, the method successfully handles various aspect ratios and resolutions while using Stable Diffusion 1.5 (pretrained at 512×512) as the base model.
> - "why optimizing similarity in this latent space helps enforce actual similarity". Essentially, the approach can be viewed as compressesing a compact representation that filters out irrelevant textural details. Specifically, Stable Diffusion's 4-channel VAE, optimized under MSE+LPIPS+GAN loss, functions as a perceptual-oriented lossy compression process rather than simple dimensionality reduction. As discussed in the appendix (L131-L136), VAE latent space provides a compact, perceptually meaningful representation that filters semantically irrelevant information. Comparing to compressing the original pixel-level signal, compressing VAE's latent variables allows the compressed bitstream to prioritize semantically salient information and avoid wasting bits on those hard-to-code and complex textures, achieving superior rate-distortion performance across downstream machine vision applications. The results in Appendix Section B.5 and Figure 14 also confirm our statement. Besides, to address potential convergence issues from extreme values of VAE latent variables, we compress the latents after normalization. We will add this implementation detail to the Appendix.

---

> > ### Comment · Reviewer_fiP4 · 2025-08-03
> >
> > - As you say in the answer to the motivation point - traditional methods focused only on fidelity, i.e. distortion, however newer compression methods focus on fidelity (both semantic and in pixel-space) and perception (e.g., perco, ddcm, MS-ILLM...). So it's not just Diff-ICMH that focuses on both. About DDCM, I think what's missing is a clear definition of what "an intelligent task" is.
> > - Figure 1 - then what's missing here in my opinion is the clear distinction that you're talking about codecs designed for use with intelligent tasks.
> > - 5.4 - So the mAP is only for object detection? could you provide in the appendix a similar ablation for an additional task?
> > - Zero-shot method - it is unclear to me why the authors claim that zero-shot approaches would degrade performance, as DDCM (a zero-shot approach) surpasses PerCo (a fine-tuned model) on compression. If a model that does zero-shot compression achieves good results on intelligent task compared to a model trained for it (Diff-ICMH) - that would raise question on the validity of the results.
> > - If you choose to discuss and report speed, and display a table of timing, then the comparison must be made on equal grounds. This is especially important since you mention complexity as the only limitation.

---

> ### Author Response · Authors · 2025-08-04
>
> ***Thank you for your valuable feedback. Here are our responses to your comments.***
> ***
> **1. Definition of "intelligent task"**
>
> We sincerely apologize for any misunderstanding caused by our unclear expression. "Intelligent task" [1] here specifically refers to neural network-based image understanding tasks, also known as "machine vision," [2,3] including object detection, instance segmentation, and pose estimation.
>
> **2. Distinction of codecs designed for use with intelligent tasks**
>
> The core differences in Figure 1 lie in the generalization capability of the codecs and the adaptability of the bitstreams to diverse downstream intelligent tasks and human visual perception needs. In subfigure (a), traditional codec-based methods employ a unified encoder-decoder for different tasks but require task-specific bitstreams due to targeted bit allocation or quantization for specific task. In subfigure (b), end-to-end neural codec methods require separate optimization of encoder-decoders for different intelligent tasks, necessitating task-specific codecs and bitstreams. In subfigure (c), feature compression-based methods require training separate codecs for compressing features of different tasks, making both the codec and bitstream task-dependent. In subfigure (d), representing our proposed Diff-ICMH, one unified codec and the corresponding bitstream are needed to meet different downstream requirements, encompassing both various intelligent tasks and human visual perception.
>
> **3. Results of ablation study**
>
>  Yes, the mAP in the ablation study was conducted on the object detection task. Thank you for your suggestion. We provide additional ablation study results for Instance Segmentation and Pose Estimation here. Results will also be incorporated into the Appendix.
>
> *Instance Segmentation:*
>
> | Method | bpp | mAP |
> |--------|-----|-----|
> | w/o semantic loss and w/o TGM | 0.0429 | 22.79 |
> | | 0.0672 | 26.37 |
> | | 0.1122 | 30.19 |
> | | 0.1572 | 31.79 |
> | w/ semantic loss and w/o TGM | 0.0347 | 23.11 |
> | | 0.0646 | 27.19 |
> | | 0.1087 | 30.53 |
> | | 0.1537 | 31.98 |
> | w/o semantic loss and w/ TGM | 0.0431 | 25.11 |
> | | 0.0658 | 27.62 |
> | | 0.1137 | 31.22 |
> | | 0.1543 | 32.21 |
> | w/ semantic loss and w/ TGM | 0.0345 | 24.86 |
> | | 0.0628 | 27.92 |
> | | 0.1059 | 31.27 |
> | | 0.1505 | 32.67 |
>
> *Pose Estimation:*
>
> | Method | bpp | mAP |
> |--------|-----|-----|
> | w/o semantic loss and w/o TGM | 0.0429 | 30.62 |
> | | 0.0672 | 41.79 |
> | | 0.1122 | 48.98 |
> | | 0.1572 | 52.59 |
> | w/ semantic loss and w/o TGM | 0.0347 | 30.66 |
> | | 0.0646 | 42.33 |
> | | 0.1087 | 49.37 |
> | | 0.1537 | 53.08 |
> | w/o semantic loss and w/ TGM | 0.0431 | 34.12 |
> | | 0.0658 | 43.27 |
> | | 0.1137 | 50.21 |
> | | 0.1543 | 53.54 |
> | w/ semantic loss and w/ TGM | 0.0345 | 33.78 |
> | | 0.0628 | 44.29 |
> | | 0.1059 | 50.46 |
> | | 0.1505 | 53.79 |
>
> **4. Discussion with zero-shot compression methods**
>
>  We apologize for the ambiguity in our earlier response. The "trained" methods we referenced specifically refer to neural codecs based on VAE architectures that are trained from scratch. We acknowledge that DDCM is indeed a solid training-free and zero-shot compression method based on diffusion models. However, the focus in this paper is on the generalization capability of codecs and bitstreams—namely, whether a single established codec and bitstream can adapt to different intelligent tasks and human perceptual needs without task-specific modifications. Under this scenario, DDCM, Diff-ICMH, and PerCo all represent generalized methods that do not require adaptive optimization for specific tasks (such as detection and segmentation), while methods like TransTIC [2] and Adapter-ICMH [3] require task-specific adaptive training to obtain specialized codecs and bitstreams. We will cite DDCM and provide clearer exposition of this categorization in the revised paper.
>
> **5. Complexity**
>
> Thank you for your valuable suggestion. The computational complexity of different methods is illustrated below. All experiments are conducted on the Kodak dataset using an A100 GPU with NFE=20 for PerCo, DiffEIC, and Diff-ICMH. Timing is measured in seconds.
> | Method    | Enc     | Dec   |
> |-----------|---------|-------|
> | VVC       | 13.862  | 0.066 |
> | ELIC      | 0.052   | 0.068 |
> | HiFiC | 0.029   | 0.050 |
> | MS-ILLM   | 0.032   | 0.051 |
> | PerCo     | 0.08    | 0.665 |
> | DiffEIC   | 0.103   | 1.329 |
> | Diff-ICMH | 0.232   | 5.456 |
>
> [1] Feng R, Jin X, Guo Z, et al. Image coding for machines with omnipotent feature learning[C]. In ECCV, 2022.
>
> [2] Chen Y H, Weng Y C, Kao C H, et al. Transtic: Transferring transformer-based image compression from human perception to machine perception[C]. In ICCV, 2023.
>
> [3] Li H, Li S, Ding S, et al. Image compression for machine and human vision with spatial-frequency adaptation[C]. In ECCV, 2024.
> ***
> ***Thank you for your valuable comments to our paper. Your constructive suggestions are greatly appreciated.***

---

> > ### Comment · Reviewer_fiP4 · 2025-08-05
> >
> > I understand now your point with regards to zero-shot methods. I still do not agree about the current usefulness of figure 1, and would suggest at least making it smaller so more crucial information such as visualization would fit.
> >
> > All of my questions were answered; I will keep an eye on further discussion with the other reviewers and will consider raising my score.

---

> > > ### Author Response · Authors · 2025-08-06
> > >
> > > Thank you very much for considering raising your score. We greatly appreciate your constructive feedback throughout the review process.

---

### Official Review · Reviewer_vy5U · 2025-06-26

**Clarity:** 3
**Significance:** 3
**Originality:** 2
**Rating:** 4
**Confidence:** 4

**Summary:**

This paper proposes a diffusion-based ICMH framework, which is implemented by a control-net and proposed *TGM*. The network is trained by rate-distortion loss, diffusion loss and proposed *SC loss*. The main contribution is that the proposed framework reserves both semantic and perceptual information, appending the generated information to obtain better down-stream task performance. This method achieves SOTA performance across various downstream tasks.

**Questions:**

- Could you provide more ablation studies evaluated on FID and LPIPS metrics ?
- Why do you adopt the middle feature in the diffusion model to calculate the *SC Loss* ? The network architecture of diffusion models should be a U-net. The middle feature of a U-net only contains the lowest scale information, which is clearly NOT like the bottleneck of an autoencoder that contains full compressed semantic information.

**Ethical Concerns:**

["NO or VERY MINOR ethics concerns only"]

**Final Justification:**

Although the novelty appears somewhat inadequate, the experimental results show the robustness of the method. Overall, I think this is a borderline paper.

**Limitations:**

No discussion on limitations.

**Paper Formatting Concerns:**

No major formatting issues.

**Quality:**

3

**Strengths And Weaknesses:**

### Strengths

- The motivation is clear and convincing.
- This paper is well experimented with ablation studies on proposed network and SOTA comparisons.
- The proposed method achieves SOTA performance across various downstream tasks.

### Weaknesses

- The contributions presented in this paper are a little insufficient: Most of the modules in the framework (e.g., autoencoder, tag extractor and stable diffusion) are pretrained.

- Poor inference efficiency: ICMH should be a very practical application-oriented field, as it is often designed for downstream tasks. However, the reported encoding/decoding time is much longer than existing methods.

---

> ### Author Rebuttal · Authors · 2025-07-28
>
> **Rebuttal to Weaknesses**
> 1. "Most of the modules in the framework (e.g., autoencoder, tag extractor and stable diffusion) are pretrained". We sincerely apologize for any confusion. As shown in Figure 4 and described in L199-L200, the tag extractor and stable diffusion model are indeed pre-trained and frozen, but the autoencoder and control module are completely trained from scratch.
> 2. "Poor inference efficiency". Yes, you are absolutely right, and this is a very important question. We have also provided corresponding experimental results and related analysis in the Appendix Section B.4. The slow decoding speed is mainly caused by the iterative denoising process. To more clearly address this concern, we will add a dedicated Limitations section in the main text to elaborate on this problem. However, it is worth noting that the core motivation of this work is to demonstrate the significant value of Diff-ICMH in unifying human perception and machine vision. The earliest autoregressive-based codecs [1,2] also required tens of seconds to decode a single image, but later, with the development of better entropy models, decoding speed improved from tens of seconds to tens of milliseconds. In recent years, inference speed optimization algorithms for diffusion models have also been rapidly developing, with numerous model distillation [3] and inference step optimization [4,5] methods emerging continuously. We believe that in the near future, combining diffusion models for image encoding and decoding will no longer be constrained by complexity and lengthy encoding/decoding times.
>
> **Answers to questions**
> 1. "More ablation studies evaluated on FID and LPIPS metrics". Thank you for your constructive suggestions. We conducted ablation studies of LPIPS and FID respectively on CLIC2020 test set to explore the impact of our proposed SC loss and TGM (tag guidance module) on subjective quality metrics. The results are shown in the table below. It can be observed that SC loss and TGM have minimal impact on LPIPS. We speculate this is because LPIPS primarily focuses on perceptual accuracy of reconstructed images, particularly low-level features, while SC loss and TGM mainly optimizes semantic consistency, which is not directly correlated with the LPIPS metric. On the other hand, SC loss and TGM significantly help reduce FID, as they implicitly and explicitly enhance semantic information in the bitstream respectively, thereby better activating the pretrained generative priors.
> | Method | bpp | LPIPS | FID |
> |--------|-----|-------|-----|
> | w/o SC loss, w/o TGM | 0.1073 | 0.1013 | 7.439 |
> |  | 0.0319 | 0.1979 | 14.81 |
> | w/ SC loss, w/o TGM | 0.1069 | 0.0967 | 7.293 |
> |  | 0.0322 | 0.1949 | 13.95 |
> | w/o SC loss, w/ TGM | 0.1071 | 0.1007 | 6.947 |
> |  | 0.0315 | 0.1986 | 13.31 |
> | w/ SC loss, w/ TGM | 0.1066 | 0.0961 | 6.821 |
> |  | 0.0318 | 0.1966 | 12.73 |
> 2. "Why do you adopt the middle feature in the diffusion model to calculate the SC Loss ?" Thank you for your constructive feedback. This is an excellent question. We initially also believed that shallower representation spaces might preserve more comprehensive information, which would be more beneficial for model training and supporting subsequent intelligent analysis. However, as shown in Figure 8(b), we found that computing the SC loss only at the middle features yields the best results, which we analyzed in L276-L278. Overall, this is likely because deeper features, i.e., middle block features, can capture higher-level semantic information while filtering out some noisy low-level perceptual information. Computing the SC loss in this representation space is more conducive to preserving high-level semantics while filtering out irrelevant information that should not be compressed and transmitted, thereby achieving better rate-distortion performance.
>
> ​[1] Cheng Z, Sun H, Takeuchi M, et al. Learned image compression with discretized gaussian mixture likelihoods and attention modules[C]//Proceedings of the IEEE/CVF conference on computer vision and pattern recognition. 2020: 7939-7948.
>
> [2] Minnen D, Ballé J, Toderici G D. Joint autoregressive and hierarchical priors for learned image compression[J]. Advances in neural information processing systems, 2018, 31.
>
> [3] Sauer A, Lorenz D, Blattmann A, et al. Adversarial diffusion distillation[C]//European Conference on Computer Vision. Cham: Springer Nature Switzerland, 2024: 87-103.
>
> [4] Sauer A, Boesel F, Dockhorn T, et al. Fast high-resolution image synthesis with latent adversarial diffusion distillation[C]//SIGGRAPH Asia 2024 Conference Papers. 2024: 1-11.
>
> [5] Meng C, Rombach R, Gao R, et al. On distillation of guided diffusion models[C]//Proceedings of the IEEE/CVF conference on computer vision and pattern recognition. 2023: 14297-14306.

---

> > ### Comment · Reviewer_vy5U · 2025-08-06
> >
> > Thanks for the author's reply. I decide to keep my score.

---

### Official Review · Reviewer_NvM5 · 2025-06-27

**Clarity:** 3
**Significance:** 3
**Originality:** 2
**Rating:** 4
**Confidence:** 4

**Summary:**

The Diff-ICMH proposed in this paper is a generative image compression framework that utilizes a diffusion-based decoder and a semantic bootstrap module. The core claim is that Diff-ICMH can support a wide range of downstream intelligence tasks - including traditional computer vision tasks and multimodal reasoning - without the need for task-specific adaptation. To support this claim, the authors have experimented on several benchmarks, including object detection, instance segmentation, keypoint estimation, and full-view segmentation on COCO; image text retrieval on Flickr30K; finger representation understanding on RefGTA; and open vocabulary segmentation on ADE20K.

**Questions:**

See weaknesses above.

**Ethical Concerns:**

["NO or VERY MINOR ethics concerns only"]

**Final Justification:**

I'd like to thank the authors for their response. I've raised the overall score.

**Limitations:**

See weaknesses above.

**Paper Formatting Concerns:**

N.A.

**Quality:**

2

**Strengths And Weaknesses:**

Strengths:

The authors claim that Diff-ICMH supports a wide range of tasks, including both traditional computer vision tasks and multimodal reasoning, without the need for any task-specific adaptation. This is a compelling and practically valuable goal, especially in the context of compressing images for general-purpose downstream usage. The authors provide empirical evidence across several tasks (e.g., object detection, instance segmentation, keypoint estimation, and panoptic segmentation on COCO; image-text retrieval on Flickr30K; referring expression comprehension on RefGTA; and open-vocabulary segmentation on ADE20K). These evaluations span different types of downstream models and include tasks involving both vision and language.

Weaknesses and Questions:
1. Limitation: A notable limitation in this manuscript is the lack of discussion of the limitations of the proposed approach. Diffusion-based decoders are known to be computationally intensive. Could the authors provide further information on the considerations and issues regarding trade-offs in terms of inference efficiency, cost and resources. Providing quantitative metrics or qualitative insights on these aspects would allow reviewers and future readers/users to better understand the practical deployment implications of Diff-ICMH.
2. Task-Independent Support: The claim of “task-independent support” remains somewhat under-tested. First, the experiments do not indicate whether any downstream tasks or their semantics were encountered during the training or labelling phases. This raises the question of whether the model truly generalizes in a zero-point manner or benefits from task-aware conditioning implicit in the training process..
3. Tag Guidance Module (TGM): The authors have conducted experiments using a Tag Guidance Module (TGM) to improve performance at negligible bitrate cost. While this design is somehow efficient, it is still too simple—relying on static tag extraction and direct prompt injection. Why doesn’t the paper explore TGM's robustness and effectiveness under more complex semantic conditions? Why not compare it with more powerful alternatives such as CLIP-based conditioning or learned prompt optimization methods? Any of these trade-offs or comparisons could significantly enhance the paper’s quality. Without such analyses, the current TGM contribution, while practical, may be perceived more as an ad-hoc engineering choice than a principled or generalizable advance. Providing such evaluations would not only strengthen the paper but also offer clearer guidance for future researchers considering whether and how to apply TGM in their own works. Since prompt-based testing is not difficult to conduct experimentally, the authors could consider including their TGM evaluations under more complex scenarios as well as comparisons with alternative prompt-based guidance methods, depending on their available resources.
4. Generalizability: The generalizability of the model to completely unseen domains or tasks (e.g., medical imaging, real-time video comprehension, or other non-visual language settings) was not explored. Thus, while the current empirical results are encouraging, the strength of the task-independence statement would be greatly enhanced by including evaluations under unseen or out-of-distribution tasks, comparisons to a task-adaptation baseline, and explicit measurements of the cost of adaptation (e.g., training time, number of fine-tuning parameters, or latency).

---

> ### Author Rebuttal · Authors · 2025-07-28
>
> **Rebuttal to Weaknesses and Questions**
> 1. "lack of discussion of the limitations of the proposed approach". Thank you very much for your feedback. Indeed, we have included experimental results on computational complexity and a discussion of limitations in Section B.4 of the Appendix. To enhance clarity, we will reorganize this content into a dedicated section in the main text to more clearly present the limitations of our approach.
> 2. "Concerns of Task-Independent Support". We sincerely apologize for any misunderstanding. We must emphasize that the entire training process does not involve any explicit or implicit optimization targeting any downstream task. Our model is optimized solely on the LSDIR dataset using diffusion-based ε-prediction loss, rate optimization, and our proposed semantic consistency loss, then directly tested on different downstream intelligent tasks. Therefore, Diff-ICMH is a general and generalizable codec that can zero-shot support different intelligent tasks once training is completed.
> 3. "Tag Guidance Module (TGM) is simple". Thank you very much for your suggestion. This is indeed a highly valuable question for discussion. In fact, the design of TGM follows a simple yet efficient philosophy, grounded in two core design principles: **First, leveraging the strong generative prior of T2I models.** As is well known, T2I models are inherently designed to generate images conditioned on text. Therefore, choosing text as the conditioning modality is a natural and reasonable choice. Methods such as CLIP image representations or learned prompt optimization approaches would deviate from the original input format of T2I models, potentially hindering the effective activation of the model's prior knowledge. We have experimented with using the CLIP grid image features of the image as the condition instead of tags. However, we found that even with CLIP features free from any compression distortion, their performance was still inferior to that of tags. As a result, we abandoned this approach. **Second, achieving negligible bitrate overhead**. Another core design principle of TGM is encoding/decoding tags, which requires extremely minimal bitrate to compress the corresponding tags as stimulation factors for activating the generative prior. As described in the paper, each image requires approximately only 100 bits. Specifically, we utilize the RAM++ version of Recognize Anything (https://github.com/xinyu1205/recognize-anything), which has a maximum default vocabulary of 4,585 tags. We employ fixed-length encoding to compress tag IDs, requiring 13 bits per ID (with a maximum capacity of 8,192). Our statistics on COCO show that each image predicts an average of 8.7 tags under RAM++'s default settings, resulting in an average bitrate of 13×8.7=113.1 bits. In contrast, compressing text would require significantly more bits, not to mention CLIP-based representations or learned prompt optimization methods, which typically involve high-dimensional floating-point continuous representations (e.g., 2,048, 3,072, or 4,096 channels). However, the CLIP-based conditioning or learned prompt optimization methods you mentioned are indeed promising directions for future research, though they would necessarily require targeted designs to address the high bitrate requirements of such representations and how to well stimulate the generative prior, which exceeds the scope of this paper. The core motivation of our work is to demonstrate that diffusion-based perceptual coding methods can effectively unify human perception and machine vision tasks. For this motivation, our designed TGM serves as a simple and practical module.
> 4. "Concerns of Generalizability". First, this work primarily validates the effectiveness of our method through common vision benchmarks, which is consistent with how other methods (Omni-ICM[1], TransTIC[2], Adapt-ICMH[3]) conduct their testing and validation. To our knowledge, we are the first to encompass such a diverse range of vision tasks and datasets, totaling 10 intelligent tasks. Other domains such as medical imaging, real-time video comprehension, and non-visual language settings are beyond the scope of this paper, though they could be explored in future work. Additionally, we have also tested on out-of-distribution (OOD) datasets, specifically RefCOCO, corresponding to the seventh subplot in Figure 6 and the related discussion at L241, and demonstrated the generalization ability of our method on OOD data.
>
> [1] Feng R, Jin X, Guo Z, et al. Image coding for machines with omnipotent feature learning[C]//European Conference on Computer Vision. Cham: Springer Nature Switzerland, 2022: 510-528.
>
> [2] Chen Y H, Weng Y C, Kao C H, et al. Transtic: Transferring transformer-based image compression from human perception to machine perception[C]//Proceedings of the IEEE/CVF International Conference on Computer Vision. 2023: 23297-23307.
>
> [3] Li H, Li S, Ding S, et al. Image compression for machine and human vision with spatial-frequency adaptation[C]//European Conference on Computer Vision. Cham: Springer Nature Switzerland, 2024: 382-399.

---

### Official Review · Reviewer_s4Y9 · 2025-06-27

**Clarity:** 3
**Significance:** 2
**Originality:** 2
**Rating:** 4
**Confidence:** 4

**Summary:**

The authors proposed Diff-ICMH, a generative image compression framework aiming for harmonizing machine and human vision in image compression. It ensures perceptual realism by leveraging generative priors and simultaneously guarantees semantic fidelity.

**Questions:**

N/A

**Ethical Concerns:**

["NO or VERY MINOR ethics concerns only"]

**Final Justification:**

The author has resolved most of my concern.

**Limitations:**

No. The authors should include a discussion of the model’s limitations. For example, perceptual-quality-oriented compression methods may introduce hallucinated details that were not present in the original image. While this may improve visual appeal, it can be problematic in domains where accuracy and fidelity are critical—such as medical imaging or scientific analysis—where hallucinated content could lead to misleading or incorrect interpretations. Highlighting this trade-off would strengthen the paper by providing a more balanced perspective on the method’s applicability.

**Quality:**

3

**Strengths And Weaknesses:**

**Pros:**
The paper addresses an important and timely research problem. Traditional lossy image compression methods are typically designed to preserve visual fidelity for human perception or storage/transmission efficiency. However, in many real-world applications, compressed images are used as inputs for downstream vision tasks. The authors correctly identify this gap and motivate their work around optimizing compression for task performance, which is a valuable direction.

---

**Mixed:**
The proposed architecture is built upon ControlNet, and the Tag Guidance Module appears to function similarly to existing tagging models. This raises some concerns about novelty. That said, the integration of a semantic consistency loss is a noteworthy aspect that could contribute to the uniqueness of the approach.

It would be helpful if the authors elaborated on their choice of similarity metric for enforcing semantic consistency—e.g., whether they used cosine similarity, Euclidean distance, or another measure—and discussed its impact on performance.

---

**Cons:**
The empirical performance of the proposed method appears limited. As shown in Figure 6, on widely used datasets such as COCO and Flickr, the method performs on par with existing perceptual compression baselines. Furthermore, Figure 7 indicates no significant improvement in perceptual quality, which weakens the empirical claims of the paper.

Additionally, the paper brings to mind related work such as [Matsubara et al., WACV 2022](https://openaccess.thecvf.com/content/WACV2022/papers/Matsubara_Supervised_Compression_for_Resource-Constrained_Edge_Computing_Systems_WACV_2022_paper.pdf), which explores supervised compression for multiple downstream tasks using a model distillation framework. This prior work seems highly relevant, yet the current paper only compares against reconstruction-based baselines. Including comparisons with task-specific or multitask compression approaches would strengthen the experimental analysis.

---

**Recommendation:**
I believe this paper sits on the borderline for acceptance. The motivation is strong, and the proposed semantic consistency idea has potential, but concerns about novelty and limited empirical gains weaken the overall contribution. I would consider increasing my score if the rebuttal addresses these concerns effectively.

---

> ### Author Rebuttal · Authors · 2025-07-28
>
> **Rebuttal to Mixed**
>
> "It would be helpful if the authors elaborated on their choice of similarity metric for enforcing semantic consistency—e.g., whether they used cosine similarity, Euclidean distance, or another measure—and discussed its impact on performance." Thanks for your constructive suggestion. Below are the ablative experiments of the choice of distortion metric to calculate SC loss, including cosine similarity and Euclidean distance. Experiments are conducted on MS COCO object detection. We follow the default training setting for both metrics. It can be observed that the results of cosine similarity achieve a better rate-distortion trade-off. We conjecture that this is because Euclidean distance is sensitive to both the direction and magnitude of vectors, making it susceptible to being dominated by large-magnitude dimensions in the features, which introduces bias in loss computation and hinders overall feature alignment. In contrast, cosine similarity eliminates the influence of vector magnitude through normalization and focuses solely on the directional relationship between vectors, thereby better capturing structural similarities between features and achieving more balanced cross-dimensional feature alignment. We will add the experimental results and corresponding analysis to the Appendix.
> | Distance Metric | bpp | mAP |
> |-----------------|-----------|-------------|
> | Cosine            | 0.0213    | 21.89       |
> |                        | 0.0345    | 26.82       |
> |                        | 0.0628    | 30.80       |
> |                        | 0.1059    | 34.01       |
> |                        | 0.1505    | 35.66       |
> | Euclidean       | 0.0251    | 21.15       |
> |                        | 0.0412    | 26.91       |
> |                        | 0.0698    | 30.33       |
> |                        | 0.1169    | 33.72       |
> |                        | 0.1633    | 35.44       |
>
>
> **Rebuttal to Cons**
> 1. About "The empirical performance of the proposed method appears limited". First, a major motivation of this paper is to explore a codec that can uniformly adapt to diverse machine vision tasks and human visual perception requirements. Moreover, we must emphasize that Figure 6 includes ten different tasks in total. Our method only performs comparably to the second-best approach on the third and seventh tasks, while demonstrating significant advantages across all other tasks.
> 2. About "Figure 7 indicates no significant improvement in perceptual quality". First, the core objective of this paper is to develop a general codec that unifies human vision and machine vision. Therefore, the inherent goal is to achieve state-of-the-art or comparable performance simultaneously in both perceptual quality and intelligent tasks. Second, our method achieves superior perceptual quality at lower bitrates compared to other approaches, such as DiffEIC.
> 3. About relationship to "Matsubara et al., WACV 2022" and "comparisons with task-specific or multitask compression approaches". First, regarding the relationship between the two papers, the WACV paper performs adaptive optimization for multiple downstream tasks to achieve better rate-distortion performance on those specific tasks, while our work does not perform any task-specific adaptive optimization and additionally supports human visual perception, making it a generalized codec. We will incorporate the citation of this work in the main text and provide a clear clarification. Furthermore, regarding comparisons with task-specific optimized codecs, we emphasize that our evaluation includes TransTIC and Adapt-ICMH—task-specific methods specifically designed and optimized for particular tasks, as demonstrated in Figure 6 and the corresponding analysis. Notably, Diff-ICMH achieves significantly superior performance compared to these specialized approaches while requiring no task-specific adaptive optimization. This demonstrates not only the exceptional performance of our method but also its superior generalization capability and practical ease of deployment.
>
> **Feedback to Limitations**
>
> We greatly appreciate your valuable suggestions. We will add a dedicated Limitations section to comprehensively address these concerns, including the trade-offs you mentioned regarding hallucinated details and applications in medical imaging and scientific analysis.

---

> > ### Comment · Reviewer_s4Y9 · 2025-08-02
> > **Thanks for the reply**
> >
> > I will increase my score accordingly.

---

> > > ### Author Response · Authors · 2025-08-04
> > >
> > > Thank you very much for your thoughtful reconsideration and for increasing your score. We greatly appreciate your constructive feedback throughout the review process.

---

### Note · Authors · 2025-08-16

We are grateful to the reviewers for their thorough feedback and constructive suggestions. We are encouraged by the reviewers' recognition of several key strengths in our work:
- **Addresses an important and practical problem (s4Y9, NvM5, vy5U)** of unifying image compression for both machine and human vision with generative priors, without task-specific tuning.
- **Achieves strong performance (NvM5, vy5U, fiP4)** across various downstream tasks with comprehensive experimental validation and thorough ablation studies.
- **Simple yet efficient design of tag-based prompts (fiP4)** to activate generative priors in text-to-image models.

Next, we summarize the main concerns by the reviewers and how we addressed them:
- **Novelty and technical design questions (s4Y9, vy5U, fiP4)**: We clarified our core contribution of incorporating generative priors to unify compression for both machine and human vision, achieved through our novel semantic consistency loss and efficient tag guidance module.
- **Computational efficiency concerns (NvM5, vy5U, fiP4)**: We acknowledge the slower inference speed due to iterative denoising and will add a dedicated Limitations section. We note that diffusion model acceleration techniques are rapidly advancing, making this limitation temporary.
- **Generalizability and task-independence validation (NvM5)**: We emphasized that our training involves no task-specific optimization, enabling true zero-shot generalization. We demonstrated out-of-distribution performance on RefCOCO to validate generalization capabilities.

We will incorporate all experimental results and analyses into the final version. We thank all reviewers for productive discussions that addressed their initial concerns. We are grateful to Reviewer fiP4 for the detailed insights and constructive suggestions throughout the rebuttal process, which significantly helped us refine our work.

---

### Decision · Program_Chairs · 2025-09-17

**Decision:**

Accept (poster)

**Comment:**

This paper develops a framew, called Diff-ICMH, which is a generative image compression framework designed to align machine and human vision in image compression. It achieves perceptual realism through the use of generative priors while also maintaining semantic accuracy.

I thank the authors for their detailed responses and additional empirical results, which effectively addressed the reviewers’ major concerns. All reviewers are now in favor of accepting the paper. I kindly ask the authors to ensure that the paper is updated with the new results and that the reviewers’ comments are thoroughly addressed. I believe the authors have summarized the necessary revisions very clearly in their “final remarks” message.